# Head-in-Head in Linear Attention

**Shijie Mei** [1 2]   **Man Yao** [1]   **Jiabo Tong** [1 2 3]   **Bo Xu** [1]   **Guoqi Li** [1]

## Abstract

The state-transition (decay) matrix governs how fixed-size memory is updated and used, making it a core design in linear attention models. Prior work exploits rank-1 approximations to reduce the cost of constructing decay matrices, but this low-rank constraint also limits the expressive capacity. We therefore formulate decay-matrix design as an open optimization problem: maximizing expressiveness while introducing minimal additional cost. Inspired by the multi-head mechanism, we propose Head-in-Head, which introduces an additional mask matrix to structure memory partitioning and interactions within a single linear-attention head. This simple, generic, and efficient design: 1) enables a rank-$r$ approximation of the decay matrix with only a few extra parameters and 2) strengthens intra-head information interaction. We further develop mask normalization and a chunk-wise parallelization scheme to support efficient parallel training. Extensive experiments on synthetic benchmarks and language modeling tasks, together with visual analyses, show that Head-in-Head consistently improves baseline performance by enriching information diversity and strengthening intra-head interactions. Code available at: https://github.com/msj-19/Head-in-Head-Linear-Attention

## 1. Introduction

Linear attention (Katharopoulos et al., 2020) mechanisms have emerged as a promising alternative to softmax-based self-attention (Vaswani et al., 2017). The core idea is to eliminate the quadratic computational complexity caused by nonlinear computations in self-attention and replace them with kernel-based decomposition, which separates the interaction between the query ($\mathbf{Q}$) and the key ($\mathbf{K}$). This enables the Key-Value states to be precomputed and reused across different queries, reducing both inference latency and memory consumption to constant $\mathcal{O}(1)$ complexity with respect to the sequence length during decoding.

Linear attention regulates how information propagates from one step to the next through various state-transition (decay) matrices (Merrill et al., 2024; Grazzi et al., 2025). Starting with RetNet (Sun et al., 2023), which adopts a fixed-value approach, subsequent work has progressively moved toward data-dependent sparse diagonal-decay matrices (Yang et al., 2024a; Qin et al., 2024; Beck et al., 2024), and has further evolved into approaches that generate dense decay matrices with non-diagonal elements (Peng et al., 2025; Yang et al., 2025). This evolution suggests that, for linear attention with a fixed memory size, designing the state transition matrix to control memory updates is crucial.

Empirically, dense state transition matrices outperform diagonal ones, indicating that richer memory state interactions can improve performance. In the limit, linear attention achieves maximal expressive power when each entry of a dense decay matrix is generated independently from $d$ parameters. Yet this ideal incurs an $\mathcal{O}(d^3)$ parameter cost, along with prohibitive computational and storage overhead during training. Recent work commonly adopts rank-1 low-rank parameterizations to generate dense decay matrices (Yang et al., 2025). Motivated by this trade-off, we revisit an underexplored design question: How can we strike a favorable balance between parameter efficiency and expressive capacity in dense decay matrices, improving representational power while introducing minimal additional parameters and computational overhead?

Inspired by the multi-head mechanism, we propose Head-in-Head, a partitioned memory design within a single linear-attention head. We introduce an $r \times r$ mask weight matrix to partition the input dimensions and impose this block structure on the dense decay matrix, enabling selective enhancement or suppression of connections between the resulting memory state blocks while remaining within the fixed memory budget of the model. This simple design produces a rank-$r$-dense decay matrix with only $r^2 d$ additional parameters. In contrast, extending standard low-rank parameterizations from rank 1 to rank $r$ requires scaling to

[1]Institute of Automation, Chinese Academy of Sciences, Beijing, China [2]School of Future Technology, University of Chinese Academy of Science, Beijing, China [3]Zhongguancun Academy, Beijing, China. Correspondence to: Guoqi Li <guoqi.li@ia.ac.cn>.

*Proceedings of the 43$^{rd}$ International Conference on Machine Learning*, Seoul, South Korea. PMLR 306, 2026. Copyright 2026 by the author(s).

$rd^2$ parameters (Siems et al., 2025). Moreover, we enable stable and efficient parallel training via mask normalization and a chunk-wise parallelization scheme tailored to Head-in-Head. Extensive experiments on synthetic benchmarks and language modeling tasks validate the effectiveness of our method. Visualizations further reveal that the proposed structured intra-head interaction substantially diversifies the patterns of information exchange in linear attention, at only a modest additional cost. Our contributions:

- We propose a plug-and-play Head-in-Head approach for linear attention, a simple, generic, and efficient design that enhances existing low-rank approximations by incorporating a block-wise masking mechanism.

- We develop a mask normalization and chunk-wise parallel scheme compatible with masks of varying mask sizes and baseline models, enabling efficient parallel training without complex structural change.

- We conduct extensive experiments across a series of tasks and baseline models. Results show that our method delivers consistent performance gains across multiple baseline linear-attention models while introducing only a small number of additional parameters.

## 2. Preliminaries

### 2.1. Notation

We use plain-text letters (e.g., $x, y$) for scalars, bold letters for tensors, italicized symbols denote vectors (e.g., $\boldsymbol{x}_t$), while non-italicized (upright) symbols represent matrices (e.g., $\mathbf{S}_t, \mathbf{A}_t$). The tensor shapes are specified upon their first occurrence in the text.

### 2.2. Softmax Attention and Linear Attention

The softmax attention mechanism computes a set of attention weights by measuring the relevance between each query and all keys; these weights are then used to compute a weighted sum of the corresponding values, as follows:

$$\boldsymbol{q}_t, \boldsymbol{k}_t, \boldsymbol{v}_t = \boldsymbol{x}_t \mathbf{W}_Q, \boldsymbol{x}_t \mathbf{W}_K, \boldsymbol{x}_t \mathbf{W}_V, \qquad (1)$$

$$\boldsymbol{o}_t = \frac{\sum_{i=1}^t e^{\boldsymbol{q}_t \boldsymbol{k}_i^\top} \boldsymbol{v}_i}{\sum_{i=1}^t e^{\boldsymbol{q}_t \boldsymbol{k}_i^\top}}, \qquad (2)$$

where $\boldsymbol{x}_t, \boldsymbol{o}_t, \boldsymbol{q}_t, \boldsymbol{k}_t, \boldsymbol{v}_t \in \mathcal{R}^{1 \times d}$, $\mathbf{W}_Q, \mathbf{W}_K, \mathbf{W}_V \in \mathcal{R}^{d \times d}$, $d$ is the model dimension. Its computational workflow introduces two critical challenges during inference: a linearly growing Key-Value (KV) cache (Kwon et al., 2023) and decode stage latency.

Linear attention (Katharopoulos et al., 2020) schemes originate from kernelized methods, by reformulating the exponential computation in softmax attention via a kernel

decomposition of the query-key products, this formulation allows the computational complexity to be reduced by pre-computing the key-value inner products and incrementally accumulating them during sequence processing:

$$\boldsymbol{o}_t = \frac{\sum_{i=1}^t \phi(\boldsymbol{q}_t)\phi(\boldsymbol{k}_i)^\top \boldsymbol{v}_i}{\sum_{i=1}^t \phi(\boldsymbol{q}_t)\phi(\boldsymbol{k}_i)^\top}, \qquad (3)$$

where $\phi(\boldsymbol{x})$ represents the kernel function. A line of works (Schlag et al., 2021a; Mao, 2022; Qin et al., 2023a) have demonstrated that a simple identity mapping can serve as an effective kernel function, leading to $\phi(\boldsymbol{x}) = \boldsymbol{x}$, and removing the normalization term originally introduced in the denominator for numerical stability has a negligible impact on model performance, thus leading:

$$\mathbf{S}_t = \mathbf{S}_{t-1} + \boldsymbol{k}_t^\top \boldsymbol{v}_t, \qquad (4)$$

$$\boldsymbol{o}_t = \boldsymbol{q}_t \mathbf{S}_t, \qquad (5)$$

where $\mathbf{S}_t \in \mathcal{R}^{d \times d}$ represents the memory state at the $t$-th timestep, continuously update and propagate over time. The final output is obtained by querying this memory state. In contrast to softmax attention, linear attention maintains a constant state cache and latency during the decode stage.

Despite the distinct origins and developmental trajectories of linear RNNs (Martin & Cundy, 2018; Qin et al., 2023b; Peng et al., 2023a), linear transformers (Katharopoulos et al., 2020), and state space models (SSMs)(Gu et al., 2022b;a; Smith et al., 2023), prior works have shown that they can be formally unified into a common linear attention framework (Chou et al., 2024; Zhang et al., 2024), defined as follows:

$$\mathbf{S}_t = \mathbf{A}_t \mathbf{S}_{t-1} + \boldsymbol{k}_t^\top \boldsymbol{v}_t, \qquad (6)$$

where $\mathbf{A}_t \in \mathcal{R}^{d \times d}$ refers to the **state transition (decay) matrix**, which governs the interaction between memory units (rows). Its diagonal entries control the self-looping mechanism within each unit, whereas its off-diagonal entries facilitate information flow across distinct units.

Prior research on linear attentions has confirmed that $\mathbf{A}_t$ plays a crucial role in data-dependent memory update (Gers et al., 2000; Graves, 2013; Jelassi et al., 2024). Thus, the design of $\mathbf{A}_t$ lies at the core of linear attention. Specifically, it began with a simple identity mapping in the original formulation. The first major advance was the shift to fixed, diagonal matrices in models like RetNet (Sun et al., 2023) and Lightning Attention (Qin et al., 2023a; Li et al., 2025), introducing a basic form of controlled state update. Subsequent models, such as HGRN2 (Qin et al., 2024), GLA (Yang et al., 2024a), Mamba (Gu & Dao, 2024; Dao & Gu, 2024), and (Sun et al., 2024; He et al., 2025; Lu et al., 2025) continuously optimize linear attention by making the diagonal decay matrix update data-dependent or using its variant. The above designs are confined to self-loop interactions

within a unit state, off-diagonal elements in $\mathbf{A}_t$ are deliberately set to zero and do not participate in state updates. More designs, such as DeltaNet (Schlag et al., 2021b; Yang et al., 2024b), RWKV7 (Peng et al., 2025), Gated-DeltaNet (Yang et al., 2025), Longhorn (Liu et al., 2024), are proposed that employ dense decay matrices with non-diagonal elements (i.e., no element in $\mathbf{A}_t$ is intentionally set to zero; for convenience, we refer to it as dense $\mathbf{A}_t$) to enable both intra-row retention and cross-row communication.

## 3. Head-in-Head Linear Attention

### 3.1. Motivation

Compared with diagonal $\mathbf{A}_t$, dense $\mathbf{A}_t$ designs enable richer information interaction and thus improve performance. A subtle yet important detail is that in diagonal $\mathbf{A}_t$, each element is generated independently from $d$ parameters, get $\mathcal{O}(d^2)$ parameters, whereas dense $\mathbf{A}_t$ cannot be constructed this way, as it would increase the parameter complexity to $\mathcal{O}(d^3)$ and substantially raise the training memory footprint by using parallel scan algorithm (Martin & Cundy, 2018; Smith et al., 2023). Dense $\mathbf{A}_t$ is typically implemented via a diagonal plus low-rank factorization for parameter efficiency in the following form:

$$\mathbf{A}_t = \text{Diag}(\boldsymbol{\lambda}_t) + \boldsymbol{a}_t^\top \boldsymbol{b}_t, \tag{7}$$

where $\boldsymbol{\lambda}_t, \boldsymbol{a}_t, \boldsymbol{b}_t \in \mathcal{R}^{1 \times d}$, with rank 1 for the low-rank part. This formulation imposes strong structural constraints on the non-diagonal elements of dense $\mathbf{A}_t$. Therefore, the design objective of dense $\mathbf{A}_t$ is to strike a favorable trade-off between the parameter efficiency and expressive capacity of $\mathbf{A}_t$, enhancing representational power while introducing minimal additional parameters and computational overhead.

### 3.2. Method

The core idea of this work is to realize structured state updates in linear attention by partitioning $\mathbf{A}_t$. This idea is inspired by the multi-head attention mechanism, a core component of modern foundation models that is widely used in both linear attention and self-attention modules. By grouping high-dimensional representations, different heads are able to capture similar or complementary information, thereby enhancing the model's ability to interpret inputs from multiple perspectives. Focusing on a single head in the linear attention setting, a natural question then arises: can its memory state matrix $\mathbf{S}_t$ also exhibit similar multiple perspectives, or can we enrich these perspectives by explicitly partitioning the memory state matrix?

To explore these issues, we propose a simple design, within each head, we group dense $\mathbf{A}_t$ matrices and add an additional trainable parameter $m$ to each group. We name this approach "Head-in-Head":

$$\begin{bmatrix} \mathbf{S}_t^1 \\ \mathbf{S}_t^2 \end{bmatrix} = \begin{bmatrix} m_1 \mathbf{A}_t^1 & m_2 \mathbf{A}_t^2 \\ m_3 \mathbf{A}_t^3 & m_4 \mathbf{A}_t^4 \end{bmatrix} \begin{bmatrix} \mathbf{S}_t^1 \\ \mathbf{S}_t^2 \end{bmatrix} + \begin{bmatrix} \boldsymbol{k}_t^1 & \boldsymbol{k}_t^2 \end{bmatrix}^\top \boldsymbol{v}_t. \tag{8}$$

In this design, the information flow between $\mathbf{S}_t^1$ and $\mathbf{S}_t^2$ is fundamentally reconfigured. For instance, when $m_2 = m_3 = 0$, the cross-system communication is completely blocked, resulting in decoupled dynamics where each subsystem evolves independently. The final output of a linear attention head can subsequently be computed as:

$$\boldsymbol{o}_t = LA(\boldsymbol{q}_t^1, \boldsymbol{k}_t^1, \boldsymbol{v}_t) + LA(\boldsymbol{q}_t^2, \boldsymbol{k}_t^2, \boldsymbol{v}_t), \tag{9}$$

where $LA(\cdot)$ denotes the employed linear attention scheme, $\boldsymbol{q}_t, \boldsymbol{k}_t$ are partitioned group-wise to $\boldsymbol{q}_t^i, \boldsymbol{k}_t^i$ along their feature dimension. Our intra-head grouping is a generic design, and thus $LA(\cdot)$ function can adopt any existing dense $\mathbf{A}_t$ design, including DeltaNet, Gated-DeltaNet, RWKV7, LongHorn (Liu et al., 2024), MesaNet (von Oswald et al., 2025), KDA (Kimi et al., 2025) etc.

Using DeltaNet as an example, Eq. (8) is defined as:

$$\mathbf{S}_t = (\mathbf{I} - \beta_t \boldsymbol{k}_t^\top \boldsymbol{k}_t \odot \mathbf{M}_t) \mathbf{S}_{t-1} + \beta_t \boldsymbol{k}_t^\top \boldsymbol{v}_t, \tag{10}$$

where $\beta_t \in \mathcal{R}$ is the memory update learning rate in DeltaNet, $\mathbf{M}_t$ is an $r \times r$ mask matrix, $\odot$ in this paper denotes the expanded matrix broadcasting element-wise multiplication, this operation is performed by uniformly expanding the participating components to the required final dimensions, then apply element-wise multiplication. $r$ is chosen such that it divides the model dimension $d$. This design allows the feature dimension to be partitioned into $r$ groups. $\mathbf{M}_t$ is expanded (via broadcasting) to match the dimensions of the state transition matrix, enabling an element-wise multiplication between the two. More head-in-head versions of $LA(\cdot)$ are provided in Appendix B.

### 3.3. State Transition Matrix

In summary, the mask matrix $\mathbf{M}_t$ exerts selective control over the strengths of inter-head connections. This design circumvents the inherent low-rank limitation of the off-diagonal components and, with an almost negligible increase in the number of parameters, raises the potential rank of the resulting state transition matrix to a general upper bound, i.e., $r$, depending on the rank of $\mathbf{M}_t$.

In this setting, intra-head grouping can be viewed as a finer-grained analogue of multi-head attention applied specifically to the query and key projections. The overall Head-in-Head module and network architecture are illustrated in Figure 1. The network architecture employs a widely used module alternate stacking strategy (Touvron et al., 2023), with our Head-in-Head Linear Attention as the token mixer block and Swish-Gated Linear Unit (Shazeer, 2020) as the channel mixer block.

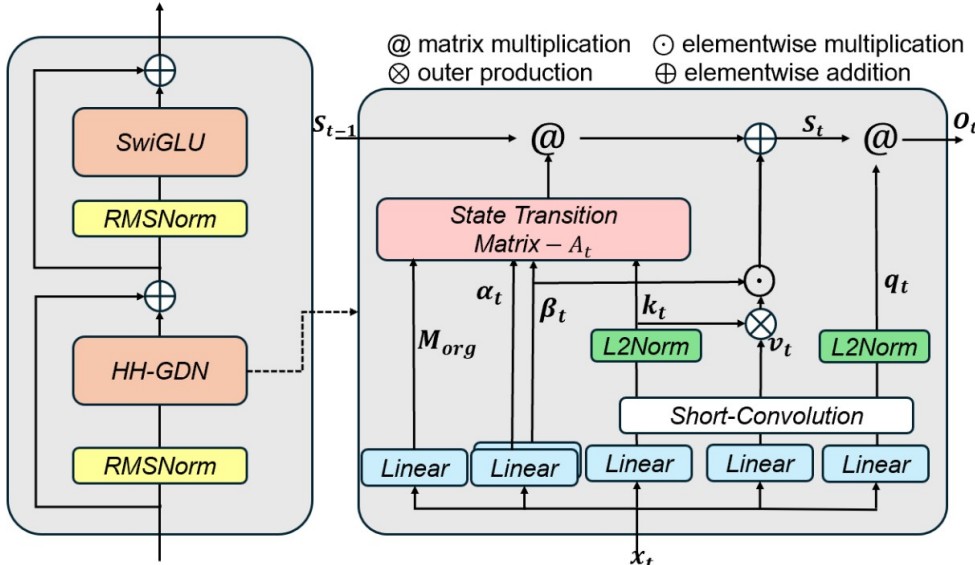

*Figure 1.* Model Architecture of Head-in-Head Linear Attention, using Head-in-Head Gated-DeltaNet (HH-GDN) for example. We use the widely used transformer architecture with Head-in-Head block as token mixer and Swish-Gated Linear Unit (SwiGLU) as channel mixer. The $\mathbf{M}_{org}$ are static learnable parameters or data-dependent mask (generated from the input via a linear projection), the number of parameters are $r^2 head\_num$ for static and $r^2 d$ for learnable which can be ignored.

### 3.4. Parallel Training Optimization

**Normalize of Mask.** Introducing an unconstrained $\mathbf{M}_t$ matrix as a selective memory-connection weight alters the state transition dynamics, which can lead to numerical instability during both training and inference, especially over long sequences. To address this, it is necessary to impose constraints that ensure the eigenvalues of the modified state-transition matrix remain within $[0, 1]$.

We disregard cases where $\mathbf{M}_t$ contains negative values, since negative sign can be absorbed into $\boldsymbol{k}_t$. We then reformulate the problem as bounding the eigenvalues of $\boldsymbol{k}_t^\top \boldsymbol{k}_t \odot \mathbf{M}_t$ within the interval $[0, 1]$, detailed reasoning is provided in Appendix A.

The matrix that meets the requirements has a large number of categories, we adopt a simple approach to generate a representative subset. For the upper bound, ensuring that all entries of the mask matrix $\mathbf{M}_t$ lie in $[0, 1]$ suffices. For the lower bound holds, we show that this is essentially equivalent to enforcing that the eigenvalues of $\mathbf{M}_t$ are non-negative, for which several control schemes exist. To achieve a natural transition from the original model with $r = 1$, we adopt the following approach: We first generate a random $r \times r$ positive mask $\mathbf{M}_{org}$ matrix and apply a L2 normalize function for each row. Subsequently, we compute $\mathbf{M}_t = \mathbf{M}_{org}\mathbf{M}_{org}^\top$.

Previous work (Grazzi et al., 2025) shows that eigenvalues in the range $[-1, 1]$ are permissible, which can be achieved by simply rescaling the original $\beta_t$ to the interval $[0, 2]$.

**Chunkwise Parallel Form for Head-in-Head Delta Networks.** Linear attention models possess both recurrent and parallel forms. The recurrent form achieves the lowest computational FLOPs, but its temporal serialization results in longer computation times due to sequential processing. In contrast, the parallel form typically incurs additional computations caused by causal mask, yet leverages GPU capabilities for single-step parallel processing to reduce computation time. To fully utilize GPU memory and accelerate operations through tensor cores, a chunk-wise computation approach is generally adopted as a balanced solution (Hua et al., 2022; Yang et al., 2024a).

Our method introduces a structured mask matrix that effectively increases the rank of interactions. This enhancement, however, incurs additional costs. Specifically, the core matrix inversion in the WY representation (Bischof & Van Loan, 1987), which reduces to a simple scalar inversion in DeltaNet, is generalized to the inversion of an $r \times r$ matrix in our framework. Moreover, the computation requires maintaining $r$ times more WY representations in memory.

The chunk-wise parallel form for our model is:

$$\mathbf{S}_t = (\mathbf{I} - \beta_t \boldsymbol{k}_t^\top \boldsymbol{k}_t \odot \mathbf{M}_t)\mathbf{S}_{t-1} + \beta_t \boldsymbol{k}_t^\top \boldsymbol{v}_t \tag{11}$$

$$= \sum_{i=1}^{t} \Big( \prod_{j=i+1}^{t} (\mathbf{I} - \beta_j \boldsymbol{k}_j^\top \boldsymbol{k}_j \odot \mathbf{M}_j) \Big) \beta_i \boldsymbol{k}_i^\top \boldsymbol{v}_i. \tag{12}$$

Following the notation in DeltaNet, assuming the chunk size is $C$, we express the $\mathbf{K}_{[t]} := \boldsymbol{k}_{tC+1:(t+1)C+1} \in \mathcal{R}^{C \times d}$ as the key block of chunk $t$, $\boldsymbol{k}_{[t]}^i = \boldsymbol{k}_{tc+i}$ denotes the $i$-th

key vector of $t$-th chunk. The initial state of chunk $t$ as use $\mathbf{S}_{[t]} := \mathbf{S}_{[t]}^0 = \mathbf{S}_{[t-1]}^C \in \mathcal{R}^{d \times d}$. Using $\mathbf{P}_i^j = \prod_{t=i}^{j}(\mathbf{I} - \beta_t \boldsymbol{k}_t^\top \boldsymbol{k}_t \odot \mathbf{M}_t) \in \mathcal{R}^{d \times d}$, $\mathbf{H}_i^j = \sum_i^j \mathbf{P}_{t+1}^j \beta_t \boldsymbol{k}_t^\top \boldsymbol{v}_t \in \mathcal{R}^{d \times d}$, we present Eq (12) as:

$$\mathbf{S}_{[t]}^n = \mathbf{P}_{[t]}^n \mathbf{S}_{[t]}^0 + \mathbf{H}_{[t]}^n. \tag{13}$$

Due to $\mathbf{M}_t \in \mathcal{R}^{r \times r}$, $\boldsymbol{k} \in \mathcal{R}^d$ and $d \bmod r = 0$. We use $\boldsymbol{k}_1, \cdots, \boldsymbol{k}_r \in \mathcal{R}^{d/r}$ to present the origin $\boldsymbol{k}$, i.e. $(\boldsymbol{k}_t = [\boldsymbol{k}_{1t}, \boldsymbol{k}_{2t}, \cdots, \boldsymbol{k}_{rt}])$. Thus, we can leverage the WY representation method (Bischof & Van Loan, 1987):

$$\mathbf{P}_{[t]}^n = \mathbf{I} - \sum_{i=1}^n \begin{bmatrix} \boldsymbol{k}_{1[t]}^{i\top} \mathbf{W}_{1[t]}^i \\ \boldsymbol{k}_{2[t]}^{i\top} \mathbf{W}_{2[t]}^i \\ \cdots \\ \boldsymbol{k}_{r[t]}^{i\top} \mathbf{W}_{r[t]}^i \end{bmatrix} = \mathbf{I} - \sum_{i=1}^n \boldsymbol{k}_{[t]}^{i\top} \odot \mathbf{W}_{[t]}^i, \tag{14}$$

$$\mathbf{H}_{[t]}^n = \sum_{i=1}^n \begin{bmatrix} \boldsymbol{k}_{1[t]}^{i\top} \mathbf{U}_{1[t]}^i \\ \boldsymbol{k}_{2[t]}^{i\top} \mathbf{U}_{2[t]}^i \\ \cdots \\ \boldsymbol{k}_{r[t]}^{i\top} \mathbf{U}_{r[t]}^i \end{bmatrix} = \sum_{i=1}^n \boldsymbol{k}_{[t]}^{i\top} \odot \mathbf{U}_{[t]}^i, \tag{15}$$

$$\mathbf{W}_{[t]}^n = \beta_{[t]}^n (\boldsymbol{k}_{[t]}^n \odot \mathbf{M}_{[t]}^n - \sum_{i=1}^{n-1} \mathbf{M}_{\phi[t]}^{ni} \mathbf{W}_{[t]}^i), \tag{16}$$

$$\mathbf{U}_{[t]}^n = \beta_{[t]}^n (\boldsymbol{v}_{[t]}^n \odot \mathbf{1}_{r \times 1} - \sum_{i=1}^{n-1} \mathbf{M}_{\phi[t]}^{ni} \mathbf{U}_{[t]}^i), \tag{17}$$

$$\mathbf{M}_{\phi[t]}^{ni} = \mathbf{M}_{[t]}^n \odot \left[ \boldsymbol{k}_{1[t]}^n \boldsymbol{k}_{1[t]}^{i\top}, \boldsymbol{k}_{2[t]}^n \boldsymbol{k}_{2[t]}^{i\top}, \cdots, \boldsymbol{k}_{r[t]}^n \boldsymbol{k}_{r[t]}^{i\top} \right], \tag{18}$$

where $\mathbf{W}_{[t]}^i \in \mathcal{R}^{r \times d}, \mathbf{U}_{[t]}^i \in \mathcal{R}^{r \times d}, \mathbf{M}_{\phi[t]}^{ni} \in \mathcal{R}^{r \times r}, \mathbf{1}_{r \times 1}$ denotes a $r \times 1$ matrix with all elements equal to 1. Then based on Eq (13) we have:

$$\mathbf{S}_{[t]}^n = \mathbf{S}_{[t]}^0 + \sum_{i=1}^n \boldsymbol{k}_{[t]}^{i\top} \odot (\mathbf{U}_{[t]}^i - \mathbf{W}_{[t]}^i \mathbf{S}_{[t]}^0), \tag{19}$$

$$\boldsymbol{o}_{[t]}^n = \boldsymbol{q}_{[t]}^n \mathbf{S}_{[t]}^0 + \sum_{i=1}^n \sum_{j=1}^r (\boldsymbol{q}_{j[t]}^n \boldsymbol{k}_{j[t]}^{i\top})(\mathbf{U}_{j[t]}^i - \mathbf{W}_{j[t]}^i \mathbf{S}_{[t]}^0). \tag{20}$$

By define $\mathbf{M}_{\phi[t]}^{ii} = \frac{1}{\beta_{[t]}^i} \mathbf{I}_{r \times r}, \mathbf{M}_{\phi[t]}^{ij} = \mathbf{0}_{r \times r}(i < j)$ additionally. We use $[\mathbf{M}_{\phi[t]}^{ij}]$ to represent the total matrix of $\mathbf{M}_{\phi[t]}$, thus:

$$\mathbf{M}_{\phi[t]} = \left[ \mathbf{M}_{\phi[t]}^{ij} \right] \in \mathcal{R}^{Cr \times Cr}, \tag{21}$$

$$\mathbf{T}_{[t]} = \mathrm{Diag}(\beta_{[t]})(\beta_{[t]} \mathbf{M}_{\phi[t]})^{-1} \in \mathcal{R}^{Cr \times Cr}, \tag{22}$$

$$\mathbf{U}_{[t]} = \mathbf{T}_{[t]}(\mathbf{V}_{[t]} \odot \mathbf{1}_{r \times 1}), \mathbf{W}_{[t]} = \mathbf{T}_{[t]}(\mathbf{K}_{[t]} \odot \mathbf{M}_{[t]}), \tag{23}$$

where $\mathbf{W}_{[t]} \in \mathcal{R}^{Cr \times d}, \mathbf{U}_{[t]} \in \mathcal{R}^{Cr \times d}$. Then we can use

chunk-level recurrent formula as:

$$\mathbf{S}_{[t+1]} = \mathbf{S}_{[t]} + [\mathbf{K}_{j[t]}^\top (\mathbf{U}_{j[t]} - \mathbf{W}_{j[t]} \mathbf{S}_{[t]})]_{j=1,\cdots,r}, \tag{24}$$

$$\mathbf{O}_{[t]} = \mathbf{Q}_{[t]} \mathbf{S}_{[t]} + \sum_{j=1}^r (\mathbf{Q}_{j[t]} \mathbf{K}_{j[t]} \odot \mathbf{L})(\mathbf{U}_{j[t]} - \mathbf{W}_{j[t]} \mathbf{S}_{[t]}), \tag{25}$$

where $\mathbf{L} \in \mathcal{R}^{C \times C}$ is the causal mask. The computational workflow here requires parallelization across the $r$ partitions. Specifically, matrix multiplications are prioritized to compute memory blocks of size $d/r \times d$, which are subsequently concatenated along the row dimension, and get the final results. Although Eq. (22) introduces a more complex inversion operation, it can still be computed with quadratic complexity in the sequence length, which matches the computational complexity of softmax attention.

# 4. Experiments

We develop the Head-in-Head mechanism on top of representative $LA(\cdot)$ architectures such as DeltaNet and Gated-DeltaNet, both of which employ rank-1 parameterizations for the non-diagonal elements. For convenience, we refer to these two variants as HH-DN and HH-GDN, respectively. We conduct comprehensive evaluations on a range of benchmarks, validating its effectiveness by synthetic benchmarks and assessing its scaling capability through standard language modeling datasets. We use $r = 4$ data-dependent mask by default.

## 4.1. Synthetic Benchmarks

On synthetic tasks, we conduct experiments on Multi-Query Associative Recall (MQAR) and Mechanistic Architecture Design (MAD) (Poli et al., 2024). The former evaluates the model's ability to retain information when handling multiple query targets, we set model dimension to 128 while the other experimental settings follow (Arora et al., 2024a). The results are presented in Figure 2. The latter (MAD) encompasses a suite of tasks designed to evaluate the architecture's capabilities in compression, information retrieval, and noise suppression. We test our model under the experimental setup outlined in (von Oswald et al., 2025), all experimental results are generated under identical configurations, with the corresponding results detailed in Table 1. Experimental results demonstrate that with the additional intra-head grouping enhancement, our model consistently outperforms its original counterpart, progressively closing the performance gap with standard self-attention mechanisms.

## 4.2. Language Modeling

We further evaluate our model's natural language modeling capabilities. Our experimental setup follows the methodol-

*Table 1.* MAD results. We keep one decimal place and round it up or down. All experiments are reproduced following the setup in (von Oswald et al., 2025).

| | In-context Recall | Noisy Recall | Fuzzy Recall | Memorize | Selection Copy | Compress | Avg. |
|---|---|---|---|---|---|---|---|
| Transformer | 91.5 | 92.9 | 57.8 | 85.0 | 100 | 54.9 | **80.3** |
| RetNet | 99.7 | 99.6 | 15.5 | 25.9 | 99.8 | 52.0 | 65.4 |
| GLA | 99.9 | 99.9 | 36.8 | 60.3 | 99.5 | 52.0 | 74.7 |
| HGRN2 | 99.9 | 99.7 | 11.5 | 86.6 | 95.7 | 53.8 | 74.6 |
| DeltaNet | 100 | 100 | 36.3 | 55.5 | 99.9 | 54.6 | 74.4 |
| GatedDeltaNet | 100 | 100 | 29.7 | 66.8 | 99.6 | 53.6 | 74.9 |
| HH-DN(ours) | 99.9 | 99.9 | 38.6 | 65.3 | 99.9 | 55.4 | 76.5 |
| HH-GDN(ours) | 99.9 | 99.9 | 41.1 | 67.0 | 99.9 | 55.3 | **77.2** |

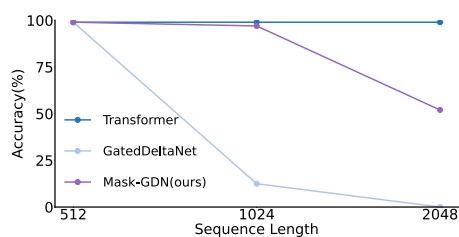

*Figure 2.* MQAR results. We keep the model dimension to 128, then training sequence from 512 to 2048.

ogy described in (Zhang et al., 2024), utilizing the SlimPajama (Soboleva et al., 2023) dataset for pre-training. We scale the experiments by training a 340M parameter model on 15B tokens and a 1.3B parameter model on 100B tokens, other training setting and datasets description can be seen in Appendix D.

**Results on commonsense reasoning tasks.** We report the perplexity (PPL) and zero-shot accuracy on commonsense reasoning tasks. These tasks take the logits of the last token as output without requiring autoregressive token generation. As shown in Table 2, our model achieves lower perplexity (PPL) and stronger performance against all models.

**Results on recall-intensive tasks.** Linear attention models are often considered weaker than self-attention in memory intensive tasks due to their fixed capacity constraints (Ballarin et al., 2024; Jelassi et al., 2024). This has motivated a series of studies aimed at enhancing the recall capabilities of linear attention. Consequently, evaluating performance on such tasks is crucial for assessing any proposed improvements. As shown in Table 3, our methods approaches the performance of self-attention models while demonstrating measurable improvements over its baseline prototype.

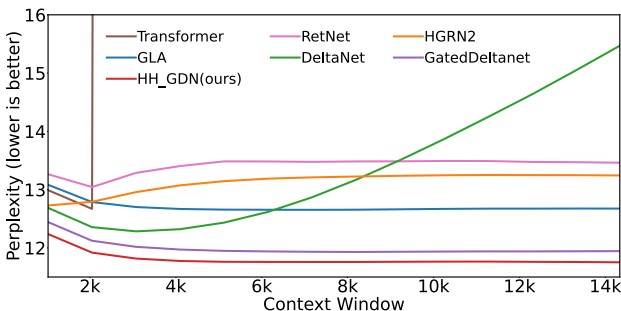

*Figure 3.* Length extrapolation. All models are 1.3B in scale, as listed in Tables 2 and 3. We evaluate them on PG-19 using the YaRN (Peng et al., 2023b) experimental code, testing sequence lengths from 1k to 16k with a step size of 1k and a sliding-window size of 256.

**Results on long context tasks.** Following (Zhang et al., 2024; Yang et al., 2025), we evaluate the model on long

context tasks LongBench (Bai et al., 2024). Table 4 illustrates the results, our method shows a certain performance improvement compared to other methods.

**Results on length extrapolation.** We evaluate the model's length generalization capability on the PG-19 dataset (Rae et al., 2019). The model is trained on sequences of length 2k and tested on sequences progressively extended up to 16k. As shown in Figure 3, with the additional intra-head grouping enhancement, our model consistently achieves lower perplexity (PPL) compared to its original baseline. Our approach does not interfere with and in fact slightly improves the length extrapolation capability provided by decay factors.

### 4.3. Analysis on Intra-head Grouping

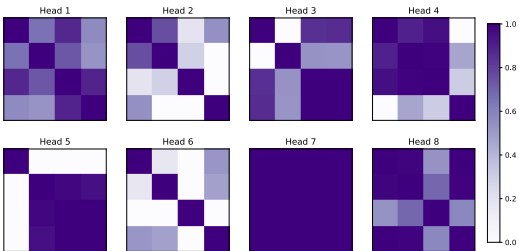

*Figure 4.* Visualization of model intra-head mask weights. We visualize the mask values from one layer of the $r = 4$ static learnable mask model to illustrate how different heads selectively attend to memory blocks. Lighter shades indicate weaker connectivity, with the absence of color representing no correlation.

We conduct ablation studies on the 1.3B-scale model to examine the effects of different $r$ and mask generation strategies, specifically comparing the performance gap between data-dependent masks and static learnable masks. The results of these evaluations are summarized in Table 5 .

Experimental results show that by increasing the number of intra-head groups, model performance improves consistently from $r = 1$ (baseline) to $r = 2$ and $r = 4$. This indicates the benefit of allowing non-diagonal elements

*Table 2.* **Results on zero-shot Common-Sense Reasoning Tasks.** [†] indicates the result is cite from (Du et al., 2025), [‡] indicates the result is obtained through open-source weights in https://huggingface.co/fla-hub. They use the same training setting with us. We conduct evaluations using lm-eval-harness (Gao et al., 2024).

| Model | Wiki. ppl↓ | Lamb. ppl↓ | ARC-e acc↑ | ARC-c acc$_n$↑ | Hella. acc$_n$↑ | Lamb. acc↑ | PIQA acc↑ | Wino. acc↑ | Avg. |
|---|---|---|---|---|---|---|---|---|---|
| *340M Params 15B Tokens L=24, d=1024* | | | | | | | | | |
| Transformer++[†] | 26.88 | 76.46 | 44.91 | **25.94** | 34.95 | 26.90 | 64.31 | 51.07 | 41.35 |
| RetNet[†] | 31.07 | 87.11 | 44.49 | 23.04 | 33.86 | 23.93 | 63.49 | 52.33 | 40.19 |
| HGRN2[†] | 27.90 | 77.40 | 45.24 | 23.63 | 35.61 | 24.74 | 65.45 | **54.06** | 41.46 |
| GLA[†] | 28.78 | 79.95 | 44.53 | 22.27 | 34.84 | 24.94 | 63.93 | 51.38 | 40.32 |
| DeltaNet | 27.72 | 71.04 | 46.13 | 25.68 | 34.90 | 24.32 | 64.69 | 51.30 | 41.17 |
| GatedDeltaNet | 26.32 | 56.03 | 46.30 | 23.55 | 35.78 | 27.36 | 65.61 | 52.17 | 41.79 |
| HH-GDN(ours) | **25.82** | **51.18** | **47.56** | 23.89 | **36.36** | **28.60** | **65.94** | 52.25 | **42.43** |
| *1.3B Params 100B Tokens L=24, d=2048* | | | | | | | | | |
| Transformer++[‡] | 17.60 | 19.32 | 54.97 | 27.82 | 49.17 | 40.77 | 70.29 | 55.56 | 49.76 |
| RetNet[‡] | 18.18 | 21.97 | 57.41 | 26.62 | 48.07 | 37.67 | 69.37 | 53.59 | 48.78 |
| HGRN2[‡] | 16.95 | 15.58 | 58.25 | 27.99 | 51.90 | 42.36 | 71.27 | 52.80 | 50.76 |
| GLA[‡] | 17.60 | 19.65 | 54.97 | 27.73 | 48.95 | 40.00 | 69.75 | 54.30 | 49.28 |
| DeltaNet[‡] | 16.72 | 15.42 | 58.54 | 26.79 | 50.24 | 42.09 | 70.51 | 52.88 | 50.18 |
| GatedDeltaNet | 16.30 | 14.52 | 57.87 | 29.01 | 52.17 | 44.36 | **72.03** | 55.09 | 51.76 |
| HH-GDN(ours) | **16.10** | **13.05** | **59.51** | 28.58 | **53.02** | **46.13** | 71.93 | **56.75** | **52.65** |

*Table 3.* **Results on Recall-Intensive Tasks.** We conduct evaluations using based-lm-eval-harness (Arora et al., 2024b).

| Model | FDA | SWDE | SQUAD | NQ | TriviaQA | Drop | Avg. |
|---|---|---|---|---|---|---|---|
| *340M Params 15B Tokens L=24, d=1024* | | | | | | | |
| Transformer++[†] | 46.14 | 25.87 | 33.22 | 18.94 | 45.97 | 20.03 | 31.70 |
| RetNet[†] | 5.90 | 9.28 | 22.41 | 6.91 | 40.05 | 18.59 | 17.19 |
| HGRN2[†] | 11.53 | 17.34 | 24.08 | 12.67 | 43.84 | 17.35 | 21.14 |
| GLA[†] | 11.26 | 16.78 | 27.85 | 12.77 | 43.90 | 17.68 | 21.71 |
| DeltaNet | **30.60** | **25.40** | 25.66 | 15.30 | 42.77 | **19.45** | 26.53 |
| GatedDeltaNet | 24.25 | 23.71 | 27.88 | 14.28 | **45.91** | 18.44 | 24.88 |
| HH-GDN(ours) | 29.16 | 24.09 | **30.47** | **15.71** | 45.56 | 18.69 | **27.28** |
| *1.3B Params 100B Tokens L=24, d=2048* | | | | | | | |
| Transformer++[‡] | 54.68 | 43.96 | 43.16 | 25.34 | 58.12 | 21.03 | 41.04 |
| RetNet[‡] | 20.07 | 26.99 | 33.46 | 16.41 | 53.14 | 19.79 | 28.31 |
| HGRN2[‡] | 13.62 | 22.68 | 32.89 | 19.54 | 55.51 | 19.26 | 27.25 |
| GLA[‡] | 27.16 | 30.08 | 34.90 | 22.20 | 55.75 | 19.36 | 31.58 |
| DeltaNet[‡] | 42.51 | 36.36 | 34.33 | 24.55 | 56.87 | 21.03 | 35.94 |
| GatedDeltaNet | 40.87 | **36.73** | 36.07 | **25.50** | 57.94 | 21.18 | 36.38 |
| HH-GDN(ours) | **46.77** | 30.36 | **37.82** | 25.47 | **58.83** | **23.14** | **37.07** |

*Table 4.* **Results on Long Bench Tasks.** We conduct evaluations using opencompass (OpenCompass, 2023) at 1.3B scale.

| Model | SQA | MQA | Sum | FS | Syn | Code | Avg. |
|---|---|---|---|---|---|---|---|
| Transformer++[‡] | 10.32 | 6.79 | 7.97 | 30.38 | 2.73 | 46.55 | 17.45 |
| RetNet[‡] | 9.10 | 5.90 | 5.44 | 20.23 | 2.49 | 41.28 | 14.07 |
| HGRN2[‡] | 8.80 | 5.94 | 7.60 | 22.48 | 1.13 | 46.80 | 15.46 |
| GLA[‡] | 9.59 | 6.09 | 7.02 | 24.90 | **3.08** | 39.86 | 15.08 |
| DeltaNet[‡] | **11.71** | 6.32 | **8.17** | 31.65 | 1.92 | 43.15 | 17.15 |
| GatedDeltaNet | 9.71 | 6.00 | 7.95 | **31.66** | 1.79 | 45.25 | 17.06 |
| HH-GDN(ours) | 9.83 | **6.89** | 7.88 | 31.17 | 1.73 | **47.94** | **17.57** |

*Table 5.* Ablation. We compare the effect of intra-head grouping enhancement on the 1.3B-scale baseline Gated-DeltaNet. First, we evaluate static learnable mask sizes by comparing $r = 2$ and $r = 4$. We further consider input-dependent mask.

| Model | Wiki.↓ | Lamb.↓ | CSR-avg | Recall-avg |
|---|---|---|---|---|
| GDN-baseline | 16.30 | 14.52 | 51.76 | 36.38 |
| +rank2 | 16.16 | 13.25 | 52.17 | 36.48 |
| +rank4 | 16.11 | 13.61 | 52.24 | **37.62** |
| +rank4+time | **16.10** | **13.05** | **52.65** | 37.07 |

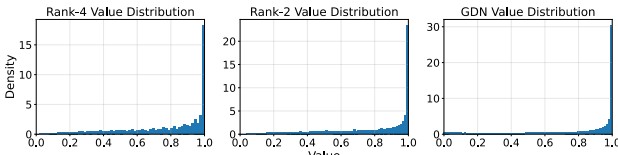

*Figure 5.* Visualization Results of All Layers Similarity. The parts less than 0 are too few and ignored, but in fact, the value range is also consistent, rank4 > rank2 > rank1.

to encode richer connectivity selectivity. While introducing data-dependent masks yields slight improvements in tasks such as perplexity reduction, it leads to a decrease in recall-oriented tasks. We therefore consider the overall impact of data-dependent masking to be marginal.

We visualize how intra-head memory selectively enhances or suppresses connectivity within a chosen layer of our model using static parameter weights (additional visualizations across more layers are provided in the appendix H). As shown in the Figure 4, the intra-head mask weights exhibit diverse patterns across different heads. Some heads retain the original rank-1 structure with all connections preserved (e.g., Head 7). Others display a clear memory block separation mechanism that suppresses cross block interaction (e.g., Head 5). Still others show more varied connectivity patterns. This diversity demonstrates the effect of our intra-head grouping enhancement, which leads to stronger and more selective memory modulation.

We empirically analyze the state distributions across three model variants: $r = 1$(GDN baseline), $r = 2$, and $r = 4$. We randomly select 20 text samples from the PG-19 dataset and truncate each to a length of 2048 tokens. For each sample, we compute and store the memory state $\mathbf{S}_t$ at every time step $t$. We then flatten $\mathbf{S}_t$ into a one-dimensional vector and measure the similarity between consecutive states using the following formula:

$$\text{Sim}(t) = \frac{\langle \text{vec}(\mathbf{S}_t), \text{vec}(\mathbf{S}_{t-1}) \rangle}{\|\text{vec}(\mathbf{S}_t)\| \cdot \|\text{vec}(\mathbf{S}_{t-1})\|}. \tag{26}$$

We aggregate the similarity values across all samples for visualization, and then compute the information entropy corresponding to the histogram as a statistical measure. The visualizations and statistical results are as Table 6 and Figure 5.

*Table 6.* Information Entropy Results. Computed by Figure 5.

| Model | Rank4 | Rank2 | Baseline |
|---|---|---|---|
| Entropy | 0.823 | 0.813 | 0.756 |

The visualizations and statistical results indicate that by introducing additional intra-head memory grouping, the

model achieves a broader distribution of memory-interaction correlations. This effectively enhances the diversity of memory selection patterns.

## 4.4. Efficiency

The experimental results demonstrate the effectiveness of our proposed modifications. To compare the inference speed of GDN, HH-GDN, and Transformer, we measure the latency and memory usage of generating 1000 new tokens under an autoregressive (token-by-token) setting at a given sequence length. The results are presented as Figure 6.

Compared to the Transformer, our model retains the inherent advantages of linear attention in decoding stage: constant latency and fixed memory usage relative to sequence length. Compared to the original Gated-DeltaNet baseline, our model maintains nearly identical memory consumption. The time-varying mask variant incurs an inference latency approximately 1.2 that of the baseline. In contrast, the fixed mask variant can pre-compute and pre-load the mask, thereby avoiding the cost of computing the mask at each step and reducing the latency overhead to only 1.05 relative to the baseline. Currently, latency and memory remain largely unchanged as the rank increases from 2 to 4.

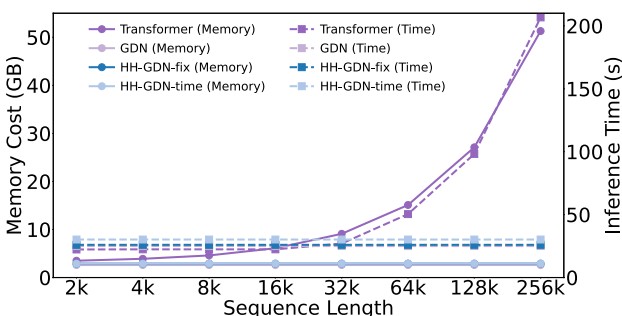

*Figure 6.* Inference time and memory cost of Transformer, Gated-DeltaNet and our HH-GDN with fixed mask and time-varying mask version. We employ the 1.3B trained model described in the preceding sections, all reported experimental results are obtained by generating an additional 1000 tokens on a single A800 GPU under fixed-length sequence conditions.

We compare the inference FLOPS and parameters in Appendix G. As for training speed, there remains room for optimization due to implementation and tuning constraints.

We provide the current benchmarking results in Appendix G.

### 4.5. Compare with Other Rank Enhanced Method

Some existing works (Siems et al., 2025) attempt to achieve more complex information propagation patterns by stacking multiple state transition matrices, but this introduces significant structural and parametric complexity. We attempted to align the parameters and compare these methods with our method in Appendix E. Our experiments demonstrate that, under equal parameter condition, our proposed method achieves better performance and inference speed.

## 5. Conclusion and Limitation

We propose Head-in-Head method, a plug-and-play enhancement for dense decay matrices in linear attention. With an additional intra-head mask and negligible parameter overhead, Head-in-Head increases the effective rank of decay-matrix generation, improving memory expressiveness and strengthening cross-row interactions. We further design an efficient parallel training scheme, building on (Tillet & Cox, 2019; Yang et al., 2025). Extensive experiments demonstrate that our method consistently boosts the expressiveness of baseline models across diverse benchmarks while preserving inference efficiency.

Our method is applicable to a broad range of existing models that incorporate non-diagonal elements in their state transition matrices. This family includes the DPLR methods (i.e. GDN (GDN-family, RWKV7, and others such as MesaNet), which can similarly benefit from rank enhancement. Currently, our primary implementation is based on the GDN-family, but we have also implemented a simple version for RWKV7. The corresponding experimental results are provided in the Appendix F.

Due to the changes introduced by increasing the rank, we must handle matrices with dimension $r$ where certain operations cannot leverage Tensor Cores, necessitating manual looping and consequently slowing down training.

### Acknowledgements

This work was partially supported by National Distinguished Young Scholars (62325603), National Science and Technology Major Project (2025ZD0215500), National Natural Science Foundation of China (62236009, U22A20103, U2541222), CAS Project for Young Scientists in Basic Research (YSBR-116), Beijing Science and 698 Technology Plan (Z241100004224011), CAAI-Tencent Rhino-Bird Open Research Fund, Beijing Zhongguancun Academy, and The Zhongguancun Academy (Grant No.s 1800302002).

## Impact Statement

This paper presents work whose goal is to advance the field of Machine Learning, and more specifically, the foundational architecture of linear attention models. There are many potential societal consequences of our work, none which we feel must be specifically highlighted here.

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

# A. Proof of Normalize.

Here we prove that through the normalization of the mask introduced in Section 3.4, we ensure the norm of the state transition matrix remains within the interval $[0, 1]$. Here we use the parameter requirements of the original DeltaNet, i.e. L2 norm for $\boldsymbol{k}_t$, $\beta_t \in [0, 1]$. Here we use $\boldsymbol{\lambda}(\mathbf{A})$ to represents the eigenvalues for matrix $\mathbf{A}$. We simply broadcast the $\mathbf{M}_t \in \mathcal{R}^{r \times r}$ to $\mathcal{R}^{d \times d}$ such that all $d/r \times d/r$ elements in each blocks equal to the corresponding single value from the original matrix.

1. $\boldsymbol{\lambda}(\mathbf{I} - \beta_t \boldsymbol{k}_t^\top \boldsymbol{k}_t \odot \mathbf{M}_t) \in [0, 1]$ equal to $\boldsymbol{\lambda}(\boldsymbol{k}_t^\top \boldsymbol{k}_t \odot \mathbf{M}_t) \in [0, 1]$.

*Proof.* Due to the fact that $\forall \beta_t \in [0, 1]$ requires eigenvalues to meet certain conditions. $\boldsymbol{\lambda}(\mathbf{I} - \beta_t \boldsymbol{k}_t^\top \boldsymbol{k}_t \odot \mathbf{M}_t) \in [0, 1]$ equal to $\boldsymbol{\lambda}(\boldsymbol{k}_t^\top \boldsymbol{k}_t \odot \mathbf{M}_t) \in [0, 1]$. □

2. Upper bound: All entries of $\mathbf{M}_t$ lie within the interval $[0, 1]$, ensure that $\boldsymbol{\lambda}(\boldsymbol{k}_t^\top \boldsymbol{k}_t \odot \mathbf{M}_t) \leq 1$.

*Proof.* We have $\boldsymbol{k}_t = [k_1, k_2, \cdots, k_d], \|\boldsymbol{k}_t\|_2 = 1$. $\forall$ each nonzero vector $\boldsymbol{x} = [x_1, x_2, \cdots, x_d] \in \mathcal{R}^{1 \times d}$, let $y_i = x_i k_i$, $\mathbf{A} = \boldsymbol{k}_t^\top \boldsymbol{k}_t \odot \mathbf{M}_t$. Then:

$$\boldsymbol{x}\mathbf{A}\boldsymbol{x}^\top = \sum_{i,j} x_i(k_i k_j M_{ij})x_j = \sum_{i,j} M_{ij} y_i y_j \leq \sum_{i,j} |y_i y_j| = (\sum_i |y_i|)^2,$$

$$\sum_i |y_i| = \sum_i |x_i k_i| \leq \|\boldsymbol{k}\|_2 \|\boldsymbol{x}\|_2 = 1 \cdot \|\boldsymbol{x}\|_2,$$

$$\boldsymbol{x}\mathbf{A}\boldsymbol{x}^\top \leq \|\boldsymbol{x}\|_2^2$$

$$\boldsymbol{\lambda}_{max}(\mathbf{A}) = Rayleigh_{max}(\mathbf{A}) = max_x \frac{\boldsymbol{x}\mathbf{A}\boldsymbol{x}^\top}{\boldsymbol{x}\boldsymbol{x}^\top} \leq 1.$$

Thus, the eigenvalues of $\boldsymbol{k}_t^\top \boldsymbol{k}_t \odot \mathbf{M}_t \leq 1$. □

3. Lower bound: $\boldsymbol{\lambda}(\boldsymbol{k}_t^\top \boldsymbol{k}_t \odot \mathbf{M}_t) \geq 0$ equal to $\boldsymbol{\lambda}(\mathbf{M}_t) \geq 0$.

*Proof.* if $\boldsymbol{\lambda}(\boldsymbol{k}_t^\top \boldsymbol{k}_t \odot \mathbf{M}_t) \geq 0$ , due to $\boldsymbol{k}_t \in \mathcal{R}^{1 \times d}$ and $\|\boldsymbol{k}_t\|_2 = 1$. then we have:

$$\boldsymbol{k}_t^\top \boldsymbol{k}_t \odot \mathbf{M}_t = Diag(\boldsymbol{k}_t)\mathbf{M}_t Diag(\boldsymbol{k}_t),$$

$$\forall \boldsymbol{x} \in \mathcal{R}^{1 \times d}, \boldsymbol{x}\boldsymbol{k}_t^\top \boldsymbol{k}_t \odot \mathbf{M}_t \boldsymbol{x}^\top = \boldsymbol{x} Diag(\boldsymbol{k}_t)\mathbf{M}_t (Diag(\boldsymbol{k}_t)\boldsymbol{x})^\top,$$

$$Rayleigh(\boldsymbol{k}_t^\top \boldsymbol{k}_t \odot \mathbf{M}_t) = \frac{x Diag(\boldsymbol{k}_t)\mathbf{M}_t (Diag(\boldsymbol{k}_t)x)^\top}{\boldsymbol{x}\boldsymbol{x}^\top} \geq \boldsymbol{\lambda}_{min}(\boldsymbol{k}_t^\top \boldsymbol{k}_t \odot \mathbf{M}_t) \geq 0.$$

Thus $\forall \boldsymbol{y} \in \mathcal{R}^{1 \times d}$ is the eigenvector of $\mathbf{M}_t$ and $\|\boldsymbol{y}\|_2 = 1$, we can have $\boldsymbol{y} = \boldsymbol{x} Diag(\boldsymbol{k}_t), \boldsymbol{x} \in \mathcal{R}^{1 \times d}$ and $\|\boldsymbol{k}_t\|_2 = 1$, therefore:

$$\boldsymbol{\lambda}(\mathbf{M}_t) = \frac{\boldsymbol{y}\mathbf{M}_t\boldsymbol{y}^\top}{\boldsymbol{y}\boldsymbol{y}^\top} = \frac{\boldsymbol{x} Diag(\boldsymbol{k}_t)\mathbf{M}_t (Diag(\boldsymbol{k}_t)\boldsymbol{x})^\top}{\boldsymbol{x}\boldsymbol{k}_t\boldsymbol{k}_t^\top \boldsymbol{x}^\top} = \frac{\boldsymbol{x} Diag(\boldsymbol{k}_t)\mathbf{M}_t (Diag(\boldsymbol{k}_t)\boldsymbol{x})^\top}{\boldsymbol{x}\boldsymbol{x}^\top},$$

$$\boldsymbol{\lambda}(\mathbf{M}_t) \geq \boldsymbol{\lambda}_{min}(\boldsymbol{k}_t^\top \boldsymbol{k}_t \odot \mathbf{M}_t) \geq 0$$

so $\forall \|\boldsymbol{k}_t\|_2 = 1$, if the condition $\boldsymbol{\lambda}(\boldsymbol{k}_t^\top \boldsymbol{k}_t \odot \mathbf{M}_t) \geq 0$ holds, then $\boldsymbol{\lambda}(\mathbf{M}_t) \geq 0$.

if $\boldsymbol{\lambda}(\mathbf{M}_t) \geq 0$, then $\forall \boldsymbol{y} \in \mathcal{R}^{1 \times d}, \boldsymbol{y}\mathbf{M}_t\boldsymbol{y}^\top \geq 0$, thus $\forall \boldsymbol{x}, \boldsymbol{k}_t \in \mathcal{R}^{1 \times d}$ and $\|\boldsymbol{k}_t\| = 1$, $\boldsymbol{x} Diag(\boldsymbol{k}_t)\mathbf{M}_t (Diag(\boldsymbol{k}_t)\boldsymbol{x})^\top \geq 0$, thus $\boldsymbol{\lambda}(\boldsymbol{k}_t^\top \boldsymbol{k}_t \odot \mathbf{M}_t) \geq 0$. Therefore $\boldsymbol{\lambda}(\boldsymbol{k}_t^\top \boldsymbol{k}_t \odot \mathbf{M}_t) \geq 0$ equal to $\boldsymbol{\lambda}(\mathbf{M}_t) \geq 0$. □

4. In order to let $\boldsymbol{\lambda}(\mathbf{M}_t) \geq 0$, there are multiple construction schemes, such as using Gershgorin circle theorem to enforce diagonal dominance. In our experiment, this scheme also produced positive performance. But it cannot naturally transition with the rank-1 structure, so we ultimately constructed it using the following scheme:

Generate a random $r \times r$ positive mask $\mathbf{M}_{org}$ matrix and apply L2 normalize function for each row. Then compute $\mathbf{M}_t = \mathbf{M}_{org}\mathbf{M}_{org}^\top$ to ensure that $\boldsymbol{\lambda}(\mathbf{M}_t) \geq 0$.

*Proof.* if $\boldsymbol{x}$ is the eigenvalue, we also have the $\mathbf{M}_{org} \in \mathcal{R}^{r \times r}$ can be broadcast to $\mathcal{R}^{d \times d}$ such that all $d/r \times d/r$ elements in each blocks equal to the corresponding single value from the original matrix by dividing a common scaling factor $\sqrt{d/r}$. Then we have

$$\boldsymbol{\lambda}(\mathbf{M}_t) = \frac{\boldsymbol{x}\mathbf{M}_t\boldsymbol{x}^\top}{\boldsymbol{x}\boldsymbol{x}^\top} = \frac{\boldsymbol{x}\mathbf{M}_{org}\mathbf{M}_{org}^\top\boldsymbol{x}^\top}{\boldsymbol{x}\boldsymbol{x}^\top d/r} = \frac{r||\boldsymbol{x}\mathbf{M}_{org}||_2^2}{d||\boldsymbol{x}||_2^2} \geq 0$$

And this method can easily regress the rank-1 structure by using the same rows in the $\mathbf{M}_{org}$. $\qquad\square$

# B. Head-in-Head in Gated DeltaNet.

Our method employs dense state-transition matrices that contain non-diagonal elements. In general, existing dense transition matrices are constructed using a diagonal-plus-low-rank scheme: the diagonal part may consist of an identity matrix, a single scalar, or a vector, while the low-rank part is formed as an outer product of two vectors, giving rise to a family of related designs. Here we give the Gated-DeltaNet version of Head-in-Head. We describe its recurrent form as:

$$\mathbf{S}_t = \alpha_t(\mathbf{I} - \beta_t \boldsymbol{k}_t^\top \boldsymbol{k}_t \odot \mathbf{M})\mathbf{S}_{t-1} + \beta_t \boldsymbol{k}_t^\top \boldsymbol{v}_t.$$

We derive the chunk-wise parallel form here:

$$\widetilde{\mathbf{P}_{[t]}^n} = \prod_{j=1}^n \alpha_{[t]}^j(\mathbf{I} - \beta_{[t]}^j \boldsymbol{k}_{[t]}^j \boldsymbol{k}_{[t]}^{j\top} \odot \mathbf{M}_{[t]}^j)$$

$$\widetilde{\mathbf{H}_{[t]}^n} = \sum_{j=1}^n \widetilde{\mathbf{P}_{[t]}^j} \beta_{[t]}^j \boldsymbol{k}_{[t]}^{j\top} \boldsymbol{v}_{[t]}^j$$

$$\mathbf{S}_{[t]}^n = \mathbf{P}_{[t]}^n \mathbf{S}_{[t]}^0 + \mathbf{H}_{[t]}^n$$

Using $\Gamma_{[t]}^{ij} = \frac{\gamma_{[t]}^i}{\gamma_{[t]}^j}, \gamma_{[t]}^i = \prod_{j=1}^i \alpha_{[t]}^j$ to express the decay products. Then $\widetilde{\mathbf{P}_{[t]}^n} = \gamma_{[t]}^n \mathbf{P}_{[t]}^n$,

$$\widetilde{\mathbf{H}_{[t]}^n} = \sum_{i=1}^n \frac{\gamma_{[t]}^n}{\gamma_{[t]}^i} \boldsymbol{k}_{[t]}^{i\top} \odot \widetilde{\mathbf{U}_{[t]}^i}, \widetilde{\mathbf{U}_{[t]}^i} = \beta_{[t]}^n(\boldsymbol{v}_{[t]}^n \odot \mathbf{1}_{r \times 1} - \sum_{i=1}^{n-1} \frac{\gamma_{[t]}^n}{\gamma_{[t]}^i} \mathbf{M}_{\phi[t]}^{ni} \widetilde{\mathbf{U}_{[t]}^i}).$$

Hence we let $\widetilde{\mathbf{M}_{\phi[t]}^{ni}} = \frac{\gamma_{[t]}^n}{\gamma_{[t]}^i} \mathbf{M}_{\phi[t]}^{ni}$,i.e.:

$$\widetilde{\mathbf{M}_{\phi[t]}^{ni}} = \frac{\gamma_{[t]}^n}{\gamma_{[t]}^i} \mathbf{M}_{[t]}^n \odot \left[\boldsymbol{k}_{1[t]}^n \boldsymbol{k}_{1[t]}^{i\top}, \boldsymbol{k}_{2[t]}^n \boldsymbol{k}_{2[t]}^{i\top}, \cdots, \boldsymbol{k}_{r[t]}^n \boldsymbol{k}_{r[t]}^{i\top}\right],$$

$$\widetilde{\mathbf{M}_{\phi[t]}^{ni}} = \Gamma_{[t]}^{ni} \mathbf{M}_{\phi[t]}^{ni}.$$

Also need define $\widetilde{\mathbf{M}_{\phi[t]}^{ii}} = \frac{1}{\beta_{[t]}^i}\mathbf{I}_{r \times r}, \widetilde{\mathbf{M}_{\phi[t]}^{ij}} = \mathbf{0}_{r \times r}(i < j)$ to get $\widetilde{\mathbf{M}_{\phi[t]}}$

$$\widetilde{\mathbf{T}_{[t]}} = \text{Diag}(\beta_{[t]})(\beta_{[t]}\widetilde{\mathbf{M}_{\phi[t]}})^{-1} \in \mathcal{R}^{Cr \times Cr},$$

$$\widetilde{\mathbf{U}_{[t]}} = \widetilde{\mathbf{T}_{[t]}}(\mathbf{V}_{[t]} \odot \mathbf{1}_{r \times 1}),$$

$$\widetilde{\mathbf{W}_{[t]}^n} = \gamma_{[t]}^n \mathbf{W}_{[t]}^n, \mathbf{W}_{[t]}^n = \mathbf{T}_{[t]}(\mathbf{K}_{[t]} \odot \mathbf{M}_{[t]}).$$

The others computation aligns to GatedDeltaNet:

$$\mathbf{S}_{[t+1]} = \widetilde{\mathbf{S}_{[t]}} + [\widetilde{\mathbf{K}_{j[t]}}^\top (\widetilde{\mathbf{U}_{j[t]}} - \widetilde{\mathbf{W}_{j[t]}}\mathbf{S}_{[t]})]_{j=1,\cdots,r},$$

$$\mathbf{O}_{[t]} = \widetilde{\mathbf{Q}_{[t]}}\mathbf{S}_{[t]} + \sum_{j=1}^r (\mathbf{Q}_{j[t]}\mathbf{K}_{j[t]} \odot \mathbf{L})(\widetilde{\mathbf{U}_{j[t]}} - \widetilde{\mathbf{W}_{j[t]}}\mathbf{S}_{[t]}),$$

where $\widetilde{\mathbf{S}_{[t]}} = \gamma_{[t]}^C \mathbf{S}_{[t]}, \widetilde{\boldsymbol{k}_{[t]}^n} = \frac{\gamma_{[t]}^C}{\gamma_{[t]}^n} \boldsymbol{k}_{[t]}^n, \widetilde{\boldsymbol{q}_{[t]}^n} = \gamma_{[t]}^n \boldsymbol{q}_{[t]}^n$

## C. Naive Recurrent Implementation

In this section, we use Python-style pseudo-code to briefly illustrate how our model operates.

```
def naive_recurrent:(q,k,v,beta,g,m,initial_state):
    '''
    q,k,v : L DK/DV
    beta,g: L
    m      : L r r #initial mask can be fixed or generated via nn.Linear(d, H*r*r)
    '''
    L, DK = q.shape
    DV = v.shape[-1]
    r = mask.shape[-1]
    o = torch.zeros_like(v)
    if initial_state is None:
        S = torch.zeros(DK, DV)
    else:
        S = initial_state
    m_org = l2norm(m.abs()) # norm for the last axis
    mask = m_org @ m_org.transpose(-1, -2) # get final mask which satisfies the previously
        established proof conditions
    q = q * (d_k ** -0.5)   # scale
    for i in range(l):
        qi,ki,vi,bi,gi,mi = map(lambda x: x[i] , (q,k,v,beta,g,mask))
        S = S * torch.exp(gi)
        ki = reshape(ki,(r,DK//r))
        wi = ki[None,:,:] * mi[:,:,None] ##r r DK//r
        wi = reshape(wi,(r,DK))
        v_new = bi*(vi[None,:]-wi@S) ##r DV
        h_sum = ki[:,None,:] * v_new[:,:,None] # r DK//r Dv
        S += reshape(h_sum,(DK,DV))
        o[i] = q@S ##DV
    return o, S
```

*Listing 1.* Pytorch-like code for naive recurrent for HH-GDN

## D. Experiments Setting.

### D.1. Synthetic Benchmarks.

For MAD and MQAR experiments we use a static learnable version to save parameters and maintain fairness, because the input-dependent learnable mask needs $nr^2d$ parameter which cannot be ignored compared to other parameters when $d$ is small (i.e. 16), while the static learnable version needs only $nr^2$ parameter. For MAD benchmark, the hyper-parameters can be found in Table 7, cite from (von Oswald et al., 2025). For MQAR tasks the hyper-parameters can be found in Table 8.

### D.2. Language Modeling.

For all models, we use SlimPajama datasets (Soboleva et al., 2023). For 340M and 1.3B models, we set context length as 2048, learning rate as 3e-4 with cosine warmup, the minimum leaning rate is 3e-5, the optimizer is AdamW (Loshchilov & Hutter, 2017) with $\beta s = (0.9, 0.95)$, weight decay 0.01,max grad norm 1.0, the other different training setting can be seen in Table 9

Regarding evaluation, all platforms used in our experiments have been listed in the main text. For generation tasks, we limit the maximum input length to 2000 characters and allow up to 48 tokens in the output.

### D.3. Datasets Description.

The commonsense reasoning tasks consist of WikiText (Merity et al., 2017), Lambada Standard (Paperno et al., 2016), ARC-Easy and ARC-challenge (Clark et al., 2018), HellaSwag (Zellers et al., 2019), PIQA (Bisk et al., 2020), WinoGrande

*Table 7.* MAD benchmark hyper-parameters.

| Hyper Parameters | Search |
|---|---|
| Dimension | 128 |
| Layers | 2 |
| Heads | 8 |
| Key Dimension | 16 |
| Training Epochs | 200 |
| Batch size | 32 |
| Learning rate Scheduler | Cosine Warmup |
|    Warm-up start learning rate | 1e-7 |
|    Warm-up steps | 0.05*Total steps |
|    minimum learning rate | 1e-5 |
| Optimizer | AdamW |
|    Learning rate | [3e-3,1e-3,5e-4,1e-4] |
|    Weight decay | [0.01,0.1] |
|    $\beta$s | (0.9,0.98) |

*Table 8.* MQAR benchmark hyper-parameters.

| Hyper Parameters | Search |
|---|---|
| Dimension | 128 |
| Layers | 2 |
| Heads | 2 |
| Value Dimension | 64 |
| Training Epochs | 64 |
| Learning rate | [np.logspace(-4,-2,4),np.logspace(-5,-3,4)] |
| Vocab size | 8192 |
| The following correspond one by one | |
| Training Length | [512,1024,2048] |
| Batch size | [128,64,64] |
| Number of KV pairs | [64,128,256] |

*Table 9.* Training Setting of Pretrain Models.

| Parameters | 340M | 1.3B |
|---|---|---|
| Layers | 24 | 24 |
| Dimension | 1024 | 2048 |
| Heads | 4 | 8 |
| Tokens | 15B | 100B |
| Total batch size | 256 | 1024 |
| Total steps | 30720 | 50016 |
| Warm up steps | 512 | 512 |
| Tied Word Embedding | True | False |

(Sakaguchi et al., 2021). Real world recall tasks consist of FDA (Arora et al., 2023), SWDE (Lockard et al., 2019), SQUAD (Rajpurkar et al., 2018), NQ (Kwiatkowski et al., 2019), TriviaQA (Joshi et al., 2017), Drop (Dua et al., 2019).

## E. Compare with Other Rank Enhanced Method.

Some existing works (Siems et al., 2025) attempt to achieve more complex information propagation patterns by stacking multiple state transition matrices, but this introduces significant structural and parametric complexity.

We compare our method with "Gated DeltaProduct", another approach for expanding the rank of the state transition matrix. Given that its token mixer does not follow the standard $4d^2$ configuration, we first fix its key-value expansion ratio to 1 to isolate the impact of state space size. We then align its rank to $r = 4$ for fair comparison. Under this setting, the standard 24-layer Gated DeltaProduct contains approximately 2B parameters.

To ensure a fair parameter comparison, we experiment with two groups of models: one group increased the number of layers in our current model to 35, reaching a 2B parameter scale, while the other group reduced the number of gated-deltaproduct layers to 17, resulting in 1.4B parameters. The comparative results are presented below.

*Table 10.* **Results on Recall-Intensive and Common-Sense Reasoning Tasks.**

| Model | Wiki. ppl↓ | Lamb. ppl↓ | ARC-e acc↑ | ARC-c $acc_n$↑ | Hella. $acc_n$↑ | Lamb. acc↑ | PIQA acc↑ | Wino. acc↑ | Avg. |
|---|---|---|---|---|---|---|---|---|---|
| *1.3B Params 100B Tokens* | | | | | | | | | |
| HH-GDN(ours) | 16.10 | 13.05 | 59.51 | 28.58 | 53.02 | 46.13 | 71.93 | 56.75 | **52.65** |
| GatedDeltaProduct | 16.23 | 13.53 | 58.08 | 29.27 | 52.20 | 45.62 | 71.98 | 55.88 | 52.17 |
| *2B Params 100B Tokens* | | | | | | | | | |
| HH-GDN(ours) | 14.96 | 11.32 | 61.28 | 30.12 | 54.34 | 48.42 | 73.39 | 58.72 | **54.38** |
| GatedDeltaProduct | 15.24 | 12.19 | 60.69 | 29.95 | 55.34 | 47.10 | 72.91 | 59.27 | 54.21 |

| Model | FDA | SWDE | SQUAD | NQ | TriviaQA | Drop | Avg. |
|---|---|---|---|---|---|---|---|
| *1.3B Params 100B Tokens* | | | | | | | |
| HH-GDN(ours) | 46.77 | 30.36 | 37.82 | 25.47 | 58.83 | 23.14 | **37.07** |
| GatedDeltaProduct | 42.42 | 30.74 | 36.04 | 25.88 | 58.00 | 22.28 | 35.89 |
| *2B Params 100B Tokens* | | | | | | | |
| HH-GDN(ours) | 53.50 | 40.21 | 38.29 | 28.10 | 61.97 | 22.66 | **40.78** |
| GatedDeltaProduct | 44.14 | 38.33 | 38.50 | 25.66 | 58.95 | 22.62 | 38.03 |

Table 10 shows the results between our method and GatedDeltaProduct at same parameter level. Our results performance better.

## F. Adaptability to Other Architectures

We completed the construction of the DPLR operator in grouped form, which means that linear attention with non diagonal elements other than MesaNet can be included as a baseline in our current work. We compared the performance with rwkv7 and the results are as follows: 340M task:

*Table 11.* Head in Head in RWKV7.

| model | Wiki.PPL↓ | Lamb.PPL↓ | Arc-e | Arc-c | Hella | Lamb | Piqa | Wino | CSR-Avg. | Recall-Avg. |
|---|---|---|---|---|---|---|---|---|---|---|
| RWKV7 | 24.67 | 33.77 | **46.97** | 24.15 | 37.34 | 31.85 | **66.05** | **52.72** | 43.18 | 29.98 |
| HH-RWKV7-r2 | **24.62** | 29.20 | 46.38 | 24.66 | 37.28 | 34.81 | 64.69 | 50.20 | 43.04 | **30.79** |
| HH-RWKV7-r4 | 24.72 | **26.67** | 46.55 | **25.17** | **37.74** | **35.77** | 65.51 | 50.75 | **43.58** | 30.74 |

## G. Inference and Train Efficiency.

We compare GDN and GatedDeltaProduct (GDP) FLOPS 12: GDP implements its recurrence by calling GDN, increasing the number of time steps by a factor of r, making the cost close to times. We ignore non-$d$ terms.

*Table 12.* Comparison of inference FLOPs between GDN and GDP.

| Stage | GDN | HH-GDN | GDP (GDN iterated r times) |
|---|---|---|---|
| Get $q, k, v, etc$ | $4d^2 + 2dH$ | $4d^2 + 2dH + dHr^2$ | $(2r + 2)d^2 + 2dH$ |
| Get $w$ | $0$ | $r \cdot (d//r) \cdot r$ (only multiply) | $0$ |
| Get $v_{\text{new}}$ | $d^2 + d$ | $r \cdot (d//r) \cdot d + rd$ | $r \cdot (d^2 + d)$ |
| Get final $\mathbf{S}$ | $d^2$ | $r \cdot (d//r) \cdot d + d^2$ | $r \cdot d^2$ |
| Get $q@\mathbf{S}$ | $d^2$ | $d^2$ | $d^2$ |
| Norm and get final $o$ | $d^2$ | $d^2$ | $d^2$ |

Ignoring non-$d^2$ terms, our method adds roughly $d^2$ extra multiplications compared to GDN, whereas GDP adds $(4r - 4)d^2$ multiplications (both addition and multiplication).

With further optimization, we expect the training cost of our method to approach that of the standard GDN implementation. For our method, we set r = 2, 4 with chunk size = 32, and r = 8 with chunk size = 16, due to the limitation of shared memory. This implies that our memory usage may be slightly smaller than the standard baseline.

The optimal speed of our current operator implementation is as follows. Using a single-layer operator with input $[B, L, D] = [1, 4096, 2048]$ and $8$ heads, the timing includes the generation of QKV projections from the input, the invocation of the attention operator, and the computation of gradients during backpropagation. This test is based on a single A800:

*Table 13.* Operator Efficiency

| | baseline-gdn | rank2 | rank4 | rank8 |
|---|---|---|---|---|
| **Memory-Cost** | 873.3 MB | 797.2 MB | 985.5 MB | 1.6 GB |
| **Time-Cost** | 6.246 ms | 8.364 ms | 11.247 ms | 16.078 ms |

The 340M parameter HH-GDN model with rank-4 completes training in approximately 17 hours on 8× A100 GPUs.

## H. Visualization Results.

Here we present all layers visualization results based on the static parameter version. The results can be seen in Figure 7,8. Some layers are characterized as all 1, consistent with the original baseline, but at the same time, more layers exhibit different feature representations. We believe that this is a more effective way to partition dimensions within a group, which helps to combine information.

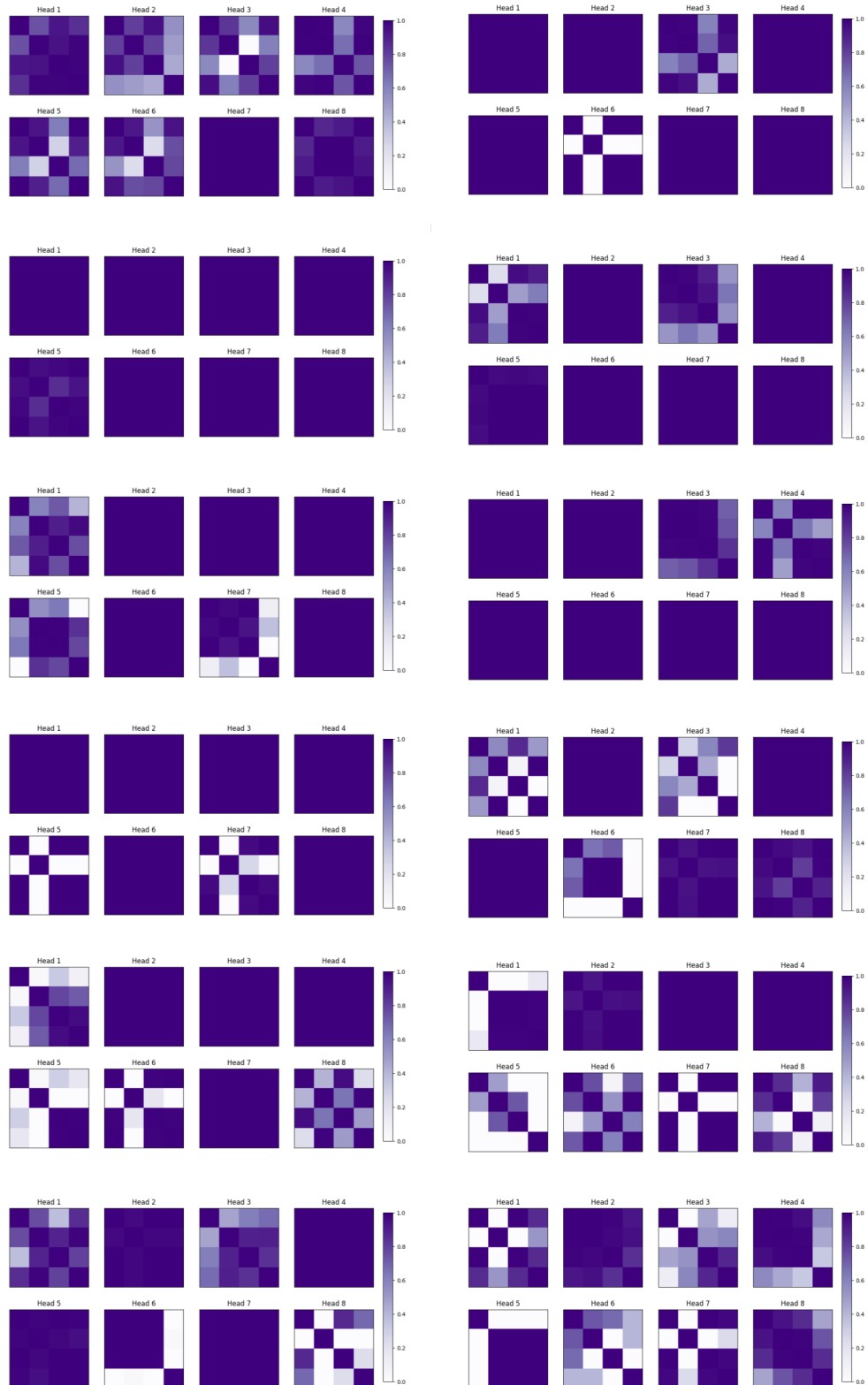

*Figure 7.* Visualization Results of Layers 1-12.

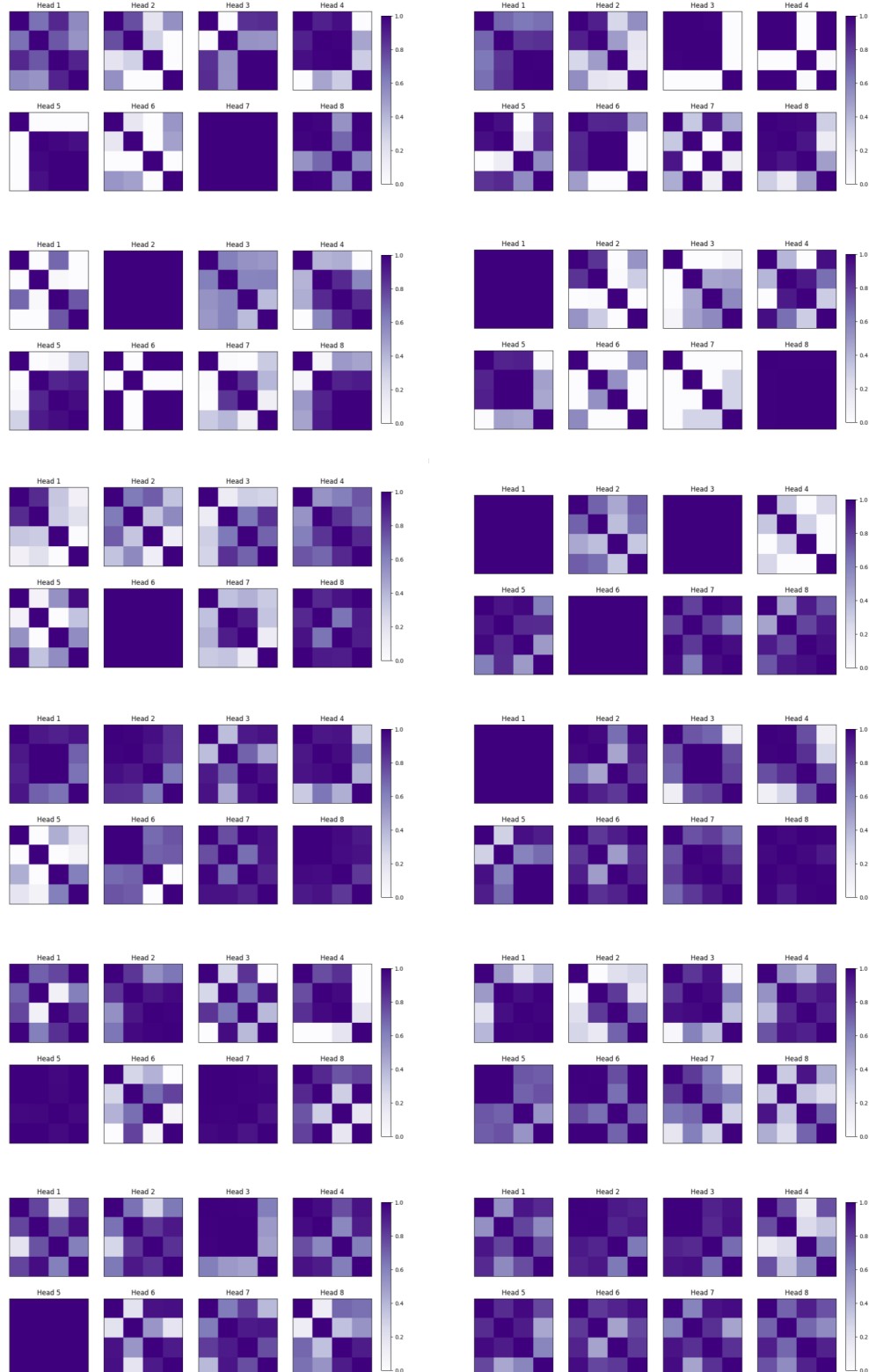

*Figure 8.* Visualization Results of Layers 13-24.

