# OpenReview forum: "Head-in-Head in Linear Attention"
_ICML.cc/2026/Conference — ICML 2026 regular_

### Official Review · Reviewer_X4Mt · 2026-03-12

**Soundness:** 3
**Presentation:** 3
**Significance:** 2
**Originality:** 2
**Overall Recommendation:** 3
**Confidence:** 4

**Summary:**

The paper introduces Head-in-Head attention, a linear attention masking strategy inspired by the multihead attention. The main idea is to partition the state matrix into r groups to increase effective rank and expressivity of the models with minimal cost. It is tested on DeltaNet and Gated DeltaNet models and it shows consistent empirical gains.

**Compliance With Llm Reviewing Policy:**

Affirmed.

**Final Justification:**

I would like to thank the authors for their detailed rebuttal. While most of my concerns are addressed, my main concern regarding the broader applicability of the method remains unsolved experimentally. Therefore, I will keep my score the same.

**Key Questions For Authors:**

1. The visualizations in appendix are very interesting as it show that some heads have the all ones structure and some layers have more of these heads than the others. Could you please discuss what this means. For example, are certain layers consistently more likely to retain rank-1-like heads than others or is the pattern mostly random? A clearer analysis here would improve the interpretability of the method and help explain when and where the added this structure is actually useful.
2. Could you explain how you initialize the M_t matrix and whether or not the performance or stability is sensitive to initialization?
3. The additional matrix operations that comes from chunking makes it unclear if the chunked formulation is actually beneficial. Can you compare the recurrent, matrix parallel, and chunkwise implementations of the same model, and report memory, throughput, and latency for each?
4. I understand that it may not be fully fair to compare Tensor Core optimized baseline implementations against a native rank r implementation of HH. However, could you also provide a comparison using native implementations for both methods, in order to give a clearer picture of the actual training throughput overhead introduced by HH?

**Limitations:**

Yes

**Strengths And Weaknesses:**

**Strengths:**
- The idea and motivation is straightforward and easy to follow.
- The paper provides a good range of experimental benchmarks.
- Shows modest but consistent improvements over the Gated Deltanet model

**Weaknesses:**
1. The paper makes very generic plug-and-play claims for dense A_t linear attention models. However, the results are limited with DeltaNet and mostly Gated DeltaNet backbones. To support this broad applicability claims, the authors should include additional compatible dense A_t backbones and compare them with their HH variants. This would clarify if the gains are specific to the DeltaNet family or general across different backbones. Additionally, it would be useful to add HH-DN results to Tables 2-4.
2. Considering the nontrivial 1.4x latency increase, and the limited experimental results, the performance gain-efficieny tradeoff and the actual applicability of the method remains unclear.
3. The ablation studies on the rank is very limited. Since the paper states that latency/memory are largely unchanged from rank 2 to 4, and there is a clear performance increase from rank 2 to 4, it raises the question if larger r yields further gains and where we can see the change in cost. So, a clear ablation study how rank affects performance, training throughput, latency and memory is needed, including greater r values. Also, analyzing the rank across model scales would provide further insights and show whether higher r is consistently beneficial.
4. The claims on the minimal overhead are rather vauge. Including the concrete additional number of parameters and FLOPs would make it more clear.
5. As there is a strong motivational similarity between GatedDeltaProduct and HH, the distinction between these to methods should be clearly differenciated with direct performance comparison in the main text, instead of the appendix. Also, the computational costs and additional parameter counts of these methods should be included in the comparison.

Additionally, I have a few minor remarks regarding the clarity and readibility of the paper:
1. There is a mistake in Equation 2, as the v_i is written as a power of exponential.
2. Since the multihead attention is the key inspiration of the paper, a dedicated small section explaining mathematically what it does, how it increases capacity would further show the relation between these two approaches.
3. While some tables have the best results highlighted, in some of them, like Table 1 and 4, the best results are not highlighted. This breaks the consistency and it gets harder to read. Additionally, Table 6 has a different format than the rest of the tables.

---

> ### Author Rebuttal · Authors · 2026-03-29
>
> Thank you for your careful reading and thoughtful evaluation.
>
> **1. On more comprehensive architectural baselines:**
> We selected GDN and DeltaNet primarily because of their well-established open-source codebases. Most dense $A_t$ methods (e.g., RWKV7, KDA, LongHorn) are mathematically similar, differing mainly in training strategies. MesaNet, require more complex DPLR iterative computations, making them difficult to implement from scratch, and their official implementations still suffer from numerical precision issues. Supporting large-scale training requires a fully functional chunk-wise operator, which is why our current work focuses on these two architectures.
>
> **2. On inference speed:**
> Linear models avoid sequence-length growth in cost. Though our latency is 1.4× baseline, it still greatly outperforms full attention on long sequences. With ongoing operator optimizations (training time already halved), we are optimistic about further acceleration. For r=2 or 4, the performance gains justify the modest overhead.
>
> **3. On comparing different values of r:**
> As shown in our response to Reviewer-`w88l`(Weakness 1)., under [H,D]=[8,256], increasing r beyond 4 yields diminishing returns relative to the substantial increase in computational cost.
>
> **4/5. Comparison with GDN and GatedDeltaProduct (GDP) FLOPS and parameters:**
> GDP implements its recurrence by calling GDN, increasing the number of time steps by a factor of r, making  the cost close to $r$ times.
>
> Regarding additional parameters and FLOPs:
>
> - For a fixed mask, our additional parameters are $Hr^2,$ which is negligible.
> - When the mask is input-dependent, the extra parameters are $HDr^2$. Compared to the $D^2$ parameters in a single linear layer for QKV projection, this is roughly $\frac{1}{64}-\frac{1}{16}$at the 1.3B scale. In contrast, GDN requires an additional $(2r-2)D^2$ parameters to generate more keys and values.
>
> For FLOPs during inference (token-mixer stage, excluding L2Norm and activation functions):
>
> | Stage| GDN|HH-GDN| GDP (GDN iterated r times) |
> | -| -| - | -|
> | Get q,k,... | $4d^2+2dH$ | $4d^2+2dH + dHr^2$| $(2r+2)d^2 + 2dH$|
> | Get w|$0$| $r*(d/r)*r$ (only multiply) |$0$|
> | Get v_new| $d^2+d$| $r*(d/r)*d + r d$ | $r*(d^2+d)$|
> | Get final_s | $d^2$| $r*(d/r)*d + d^2$| $r*d^2$|
> | Get q@S| $d^2$| $d^2$| $d^2$|
> | Get final output | $d^2$| $d^2$| $d^2$|
>
> Ignoring non-$d^2$ terms, our method adds roughly $d^2$ extra multiplications compared to GDN, whereas GDP adds $(4r-4)d^2$ multiplications (both addition and multiplication).
>
> The observed inference slowdown likely stems from suboptimal inference operator implementations and additional memory access pressure. During training, we also need to store $r$ times the WY representation and a $r \times r$ inverse matrix, which increases memory and computation.
>
> **Minor remarks (1 & 3):**
> We will correct the typos and ensure consistency across tables.
>
> **2.On multi-head attention and memory interaction:**
> Multi-head attention provides an interpretable framework for memory interaction. When we block-diagonalize so that no memory partitions interact, DeltaNet reduces to GLA. This does not affect the total memory capacity but changes the interaction strategy across memory rows. Our visualizations and empirical results indicate that richer interaction strategies improve model expressiveness. This design may also inspire further thinking: in traditional multi-head attention, allowing queries and keys from different heads to couple might yield even better performance.
>
> **Response to key questions:**
>
> **Question 1:**
> Our results show that broader mask patterns lead to stronger performance, suggesting that the model benefits from freely exploring connectivity structures in memory space. A single fixed pattern is less effective. While it is difficult to determine which layers converge to identity, we observe that some heads converge to block-diagonal patterns.
>
> **Question 2:**
> We use a simple initialization scheme. For time-invariant masks, we apply `init.kaiming_uniform_(self.mask, a=math.sqrt(5))`. For time-varying masks (i.e., using `nn.Linear`), the initialization follows the same approach as for other linear layers, with no special handling.
>
> **Question 3 (Naive implementation training speed comparison):**
> To accelerate training, we first recognized the importance for an efficient chunk-wise operator. Using a completely naive implementation with r = 4 as an example, and comparing with our current best implementation on a single A800 GPU, we tested a single layer with input shape `B H L D = [1, 8, 4096, 256]`. After warmup, the average forward+backward time and memory usage are:
>
> - Naive recurrent: 330 ms, 30.2 GB
> - Naive chunk-all: 2200 ms, 7.3 GB
> - Triton chunk-wise: 15 ms, 960 MB
>
> This demonstrates the necessity of chunk-wise training implementations beyond native PyTorch.
>
> **Question 4:**
> We have compared the training cost of our operator in our response to Reviewer-`o7ec` (Question 3).

---

> > ### Author Rebuttal · Reviewer_X4Mt · 2026-04-02
> >
> > I would like to thank the authors for their detailed rebuttal. While most of my concerns are addressed, my main concern regarding the broader applicability of the method remains unsolved experimentally. Therefore, I will keep my score the same.

---

> > > ### Author Response · Authors · 2026-04-03
> > >
> > > Thank you for your reply. We believe that the current architecture implementation already includes most of the DPLR operators.
> > >  If necessary, we will consider addressing the implementation of KDA and RWKV7 in subsequent implementations, but this will take time.
> > >  We will try to quickly iterate a version of the operator that can be used for training to solve your question.
> > >
> > > We have currently completed the construction of the DPLR operator in grouped form, which means that linear attention with non diagonal elements other than MesaNet can be included as a baseline in our current work. We compared the performance with rwkv7 and the results are as follows:
> > > 340M task:
> > >
> > > | model          | wiki.PPL↓      | lamb.PPL↓      | arc-e | arc-c | hella | lamb  | piqa  | wino  | avg             |
> > > | -------------- | --------------- | --------------- | ----- | ----- | ----- | ----- | ----- | ----- | --------------- |
> > > | baseline-GDN   | 26.29           | 58.90           | 45.79 | 24.32 | 36.05 | 26.08 | **66.27** | 50.99 | 41.58           |
> > > | HH-GDN-r2      | 25.52           | 42.37           | 45.92 | 24.40 | 36.79 | 30.29 | 65.61 | **51.46** | 42.41           |
> > > | baseline-RWKV7 | 25.06           | 34.59           | 45.88 | 23.89 | 37.14 | 31.13 | 66.10 | 51.38 | 42.58           |
> > > | HH-RWKV7-r2    | **24.62** | **29.20** | **46.38** | **24.66** | **37.28** | **34.81** | 64.69 | 50.20 | **43.04** |
> > >
> > > Training loss  can be seen in https://anonymous.4open.science/r/hh_linear_attn-3D67/loss_compare.png

---

### Official Review · Reviewer_w88L · 2026-03-13

**Soundness:** 2
**Presentation:** 3
**Significance:** 3
**Originality:** 3
**Overall Recommendation:** 3
**Confidence:** 2

**Summary:**

This paper proposes Head-in-Head, a modification for linear attention models that increases the expressive capacity of the state transition (decay) matrix used to update memory states. In many existing linear attention architectures, dense decay matrices are approximated using rank-1 parameterizations to maintain parameter efficiency. This constraint restricts the ability of the model to represent richer interactions between memory units. The proposed method partitions the memory state within a single attention head and introduces a mask matrix that controls the interaction between these partitions. As illustrated in Figure 1, this design increases the effective rank of the decay matrix while introducing only a small number of additional parameters.

The method is designed to be a plug-and-play component that can be integrated into existing linear attention architectures such as DeltaNet and Gated-DeltaNet. The paper also introduces mask normalization to maintain stability of the state transition matrix and proposes a chunk-wise parallel training scheme to preserve efficient GPU computation. Experiments are conducted on synthetic benchmarks such as MQAR and MAD (Figure 2 and Table 1), as well as language modeling tasks including commonsense reasoning, recall-intensive benchmarks, and long-context evaluation (Tables 2-4). The results show consistent performance improvements over baseline linear attention models while maintaining the inference efficiency advantages associated with linear attention architectures.

**Compliance With Llm Reviewing Policy:**

Affirmed.

**Key Questions For Authors:**

1. The experiments mainly compare the method with closely related linear attention architectures such as DeltaNet and Gated-DeltaNet. Since Transformers remain the dominant architecture for many sequence modeling tasks, it would be helpful to understand more clearly how the proposed method compares with standard Transformer baselines in terms of both performance and efficiency.

2. Table 5 suggests that performance improves when increasing the number of intra-head partitions from r = 1 to r = 4. Have the authors tested larger values of r, and if so, what limits further scaling?

3. Section 3.3 introduces a chunk-wise parallel training formulation. Can the authors clarify the practical training overhead compared with the original DeltaNet implementation?

**Limitations:**

Yes, partially. The paper includes a brief discussion of limitations in Section~5, where the authors note that increasing the rank of the decay matrix introduces additional computational overhead. In particular, the method requires operations on matrices of size $r \times r$, which cannot fully utilize Tensor Core acceleration and therefore may slow down training. The authors also acknowledge that some operations require manual looping, which further increases training time.

**Strengths And Weaknesses:**

Soundness

Strength

The technical formulation of the method is clearly specified. The paper explains how the mask matrix modifies the state transition matrix and increases the effective rank of the decay matrix. The mathematical formulation of the intra-head partition mechanism in Eq. (8) and Eq. (10) clearly describes how different partitions interact during memory updates. The stability discussion in Section 3.3 also provides a mechanism to constrain eigenvalues of the transition matrix through mask normalization.
The empirical evaluation is conducted across several types of tasks. On synthetic benchmarks such as MAD, the proposed method improves performance compared with baseline models (Table 1). The MQAR experiment in Figure 2 also shows improved recall ability when sequence length increases. The evaluation on language modeling tasks demonstrates lower perplexity and improved accuracy compared with baseline linear attention models (Table 2). The ablation study in Table 5 further analyzes the effect of increasing the number of intra-head partitions.

Weakness

The experimental improvements are generally moderate across many tasks. The comparisons mainly focus on closely related linear attention architectures such as DeltaNet and Gated-DeltaNet. The evaluation does not include comparisons with a wider range of efficient sequence models.

In addition, the experimental setup does not clearly position the proposed method relative to standard Transformer models. Although Transformer results appear in some tables (for example Table 1 and Table 2), they are not consistently used as primary baselines. Since the proposed method increases the complexity of the decay matrix and introduces additional computation, a clearer comparison against standard Transformer architectures would help assess whether the method meaningfully improves the performance-efficiency trade-off.
The efficiency trade-off introduced by the method is also not fully analyzed. Figure 6 indicates that inference latency increases compared with the baseline Gated-DeltaNet model. The paper does not provide a detailed analysis of how this cost scales with model size or with larger values of the partition parameter r.

Presentation

Strength

The paper is generally well structured and easy to follow. The motivation for improving the expressiveness of the decay matrix is clearly described in Section 3.1. The architecture diagram in Figure 1 provides a helpful overview of how the Head-in-Head mechanism is integrated into the linear attention framework. Tables summarizing benchmark results (Tables 1-4) clearly present the experimental comparisons.
The ablation analysis and visualization also help illustrate the behavior of the proposed mechanism. For example, the visualization of intra-head mask weights in Figure 4 shows how different heads learn distinct interaction patterns between memory partitions.

Weakness

Some parts of the method description are difficult to follow. Section 3 introduces several equations and derivations in rapid succession, which makes the mechanism difficult to understand without strong familiarity with DeltaNet-style architectures. The chunk-wise parallel training description in Section 3.3 is particularly dense and would benefit from additional intuitive explanation.
The terminology used to describe partitions is also somewhat inconsistent. The paper alternates between terms such as groups, blocks, and intra-head partitions. Providing a short algorithm summary or pseudocode for the method could improve clarity.

Significance

Strength

The work addresses an important problem in efficient sequence modeling. Linear attention architectures are attractive for long-sequence tasks because they avoid the quadratic complexity of softmax attention. Improving the expressiveness of the memory update mechanism is therefore an important research direction.
The experimental results indicate that the method improves performance on several evaluation settings. Improvements appear on recall-intensive benchmarks (Table 3) and long-context tasks (Table 4). The analysis of memory state similarity in Figure 5 also suggests that the proposed partitioning mechanism increases the diversity of memory interaction patterns.

Weakness

The magnitude of improvements across benchmarks is relatively small in several cases. It remains unclear whether the proposed modification would significantly change the practical performance of large-scale models or production systems. The evaluation also focuses mainly on moderate-scale models (340M and 1.3B parameters), so the scalability of the approach to larger models is uncertain.

Originality

Strength

The paper introduces a new architectural mechanism that increases the effective rank of the decay matrix by partitioning memory within a single attention head. While the idea is conceptually inspired by multi-head attention, applying this idea inside a single head through a structured mask matrix appears to be a novel design choice.
The visualization results in Figure 4 further illustrate how the mask matrix enables different connectivity patterns between memory partitions, supporting the claim that the model learns richer memory interaction structures.

Weakness

The method builds directly on existing linear attention architectures such as DeltaNet and Gated-DeltaNet. The contribution mainly modifies the parameterization of the state transition matrix rather than introducing a new modeling framework. The novelty therefore lies primarily in architectural design rather than in a fundamentally new theoretical formulation.

---

> ### Author Rebuttal · Authors · 2026-03-29
>
> Thank you for your thoughtful review and detailed comments.
>
> **Response to weakness:**
>
> 1.**Comparison with a broader range of architectures**:
> Our choice of baselines was primarily guided by model performance and the availability of open-source implementations within the community. Adopting a consistent training pipeline allowed us to minimize the effort required to reproduce prior work. The open-source baselines included in the FLA framework have been largely incorporated into our comparisons. Given that we conducted experiments on Gated-DeltaNet (GDN), we retrained the 1.3B GDN model, for its model was not publicly available.
>
> Since our design is intended primarily for linear attention architectures, we did not use full-attention models as primary baselines due to their well-known inefficiency on long sequences. Instead, we report the performance of Transformers at different scales and highlight the advantages of linear attention architectures on extremely long sequences in Figure 6. We note that architectures combining linear attention and full attention are emerging as a growing trend, and thus we believe our main focus should be on comparing with existing linear attention models rather than with full-attention ones.
>
> **Tradeoff for different r**:
> We provided a detailed description of our current training speed in question 3 of review-`o7ec`
> To compare the performance of different ranks, we set give a 340M experience here.
> | model| csr-avg | recall-avg |
> | - | - | - |
> | baseline-gdn | 41.58 | 26.63|
> | r2| 42.41   | 26.22|
> | r4| **42.43**   | **27.28**|
> | r8| 42.42| 27.14 |
>
>
>
> 2.**Clarity of method description**:
> We will include the full DeltaNet derivation in the appendix and clarify terminology: “block” refers to matrix blocks, “group” to memory partitions within each head. Pseudocode will be provided in the supplementary material (see below) to improve clarity.
>
> ```python
> def naive_recurrent(q, k, v, beta, g, mask, initial_state):
>     '''
>     q, k, v: L DK / DV
>     beta, g: L
>     mask: L r r  # already normalized to obtain final mask
>     '''
>     L, DK = q.shape
>     DV = v.shape[-1]
>     r = mask.shape[-1]
>     o = torch.zeros_like(v)
>     if initial_state is None:
>         S = torch.zeros(DK, DV, device=k.device, dtype=torch.float32)
>     else:
>         S = initial_state
>     q = q * (d_k ** -0.5)  # scale
>     for i in range(l):
>         qi,ki,vi,bi,gi,mi = map(lambda x: x[i] , (q,k,v,beta,g,mask))
>  	S = S * torch.exp(gi)
> 	ki = reshape(ki,(r,DK//r))
> 	wi = ki[None,:,:] * mi[:,:,None] ##r r DK//r
> 	wi = reshape(wi,(r,DK))
> 	v_new = bi*(vi[None,:]-wi@S) ##r DV
> 	h_sum = ki[:,None,:] * v_new[:,:,None] # r DK//r Dv
> 	S += reshape(h_sum,(DK,DV))
>         o[i] = q@S ##DV
>     return o, S
> ```
>
> 3.**Larger-scale experiments**:
> We acknowledge that experiments at larger scales are challenging given our limited resources. Nevertheless, our results show consistent performance improvements, and we believe the advantages of our method are well demonstrated at the current pretraining scales. In the supplementary material, we have included results at the 2B scale with 100B tokens, where our method continues to outperform Gate_DeltaProduct at comparable parameter counts, and also shows improved scalability compared to the 1.3B models.  In particular, at the 1.3B scale, we observe clear and consistent performance gains across a diverse set of tasks, demonstrating the effectiveness and generalizability of our method beyond smaller-scale validation.
>
> 4.**Originality**: You can consider our work as a supplement to the existing state transition matrix containing non diagonal elements.
>
> **Response to key questions:**
>
> 1. We have clarified the rationale behind our choice of baselines in weakness 1.
> 2. We report results from rank1-8 in weakness 1. We do not use larger r values because the current operator implementation requires solving a matrix inversion of size `(BT r) × (BT r)` and storing an additional factor of r times the WY representation. Loading all of these simultaneously can exceed the shared memory limits of the GPU. Moreover, as r increases, the performance gains become marginal compared to the substantial increase in training cost. We selected r = 4 because it offers a favorable trade-off between acceptable training overhead and strong performance.
> 3. To accelerate training, we first recognized the need for an efficient chunk-wise operator, as adopted in models such as GLA and DeltaNet. Using a completely naive implementation with r = 4 as an example, and comparing with our current best implementation on a single A800 GPU, we tested a single layer with input shape `B H L D = [1, 8, 4096, 256]`. After warmup, the average forward+backward time and memory usage are:
>
> - Naive recurrent: 330 ms, 30.2 GB
> - Naive chunk-all: 2200 ms, 7.3 GB
> - Triton chunk-wise: 15 ms, 960 MB
>
> Training time in review-`o7ec` **question 3 Trade-offs and Inference Time**

---

> > ### Author Rebuttal · Reviewer_w88L · 2026-04-04
> >
> > Most of my concerns have been well addressed. However, the applicability and scalability remains an unresolved concern to me. I will make my final scoring decision after considering the feedback of the other reviewers.

---

> > > ### Author Response · Authors · 2026-04-08
> > >
> > > Thank you for your reply. Regarding the scalability of the model, we currently need 32 A graphics cards to train the 1.3B model for about 4 days. To complete a larger scale model that can be compared, we need to train the comparison baseline at the same time. Currently, we do not have so many resources.
> > > Regarding the adaptability of the model, we spent a lot of time during the rebuttal period to complete the operator implementation of the general DPLR on our method. Currently, the model is being trained, and from the perspective of training trends, it can ensure the scalability of the algorithm in different implementations.
> > > https://anonymous.4open.science/r/hh_linear_attn-3D67/loss_compare.png
> > > More results have been added into the rebuttal comment in review `X4Mt`.

---

### Official Review · Reviewer_o7ec · 2026-03-15

**Soundness:** 2
**Presentation:** 2
**Significance:** 3
**Originality:** 3
**Overall Recommendation:** 4
**Confidence:** 4

**Summary:**

This paper proposes Head-in-Head, a lightweight modification to dense-decay linear attention that partitions a single head into multiple intra-head groups and uses a blockwise mask to enrich state interactions within the decay matrix. The method is mainly instantiated on DeltaNet and Gated-DeltaNet, together with mask normalization and a chunk-wise parallelization scheme, and is evaluated on synthetic tasks, language modeling, recall-intensive benchmarks, and efficiency analysis. Overall, the paper presents an interesting and reasonably motivated architectural refinement, but the gains are mostly incremental and some claims feel slightly stronger than what the evidence directly establishes.

**Compliance With Llm Reviewing Policy:**

Affirmed.

**Final Justification:**

The rebuttal has helped clarify my main technical concern regarding the spectral argument, and the additional rank-based analysis makes the proposed mechanism more convincing than before. That said, I still find the evidence for the underlying mechanism somewhat indirect, and I believe the paper would be further strengthened by a clearer causal analysis of why the method leads to the observed improvements. Overall, however, I now view the work as technically sound and the contribution as meaningful, and I am therefore inclined to raise my score.

**Key Questions For Authors:**

1. **Presentation and organization of tables, figures, and equations.**
The paper would benefit from a more carefully organized presentation. In particular, the placement of Tables 1--4 and Figures 2--4 currently feels somewhat unstructured, which makes the empirical narrative harder to follow than necessary. I also have several specific presentation questions: in Table 3, should the 340M setup use $L=24$ rather than $L=12$? In Table 4, why is only the Avg column boldfaced while other best-performing entries are not highlighted consistently? In addition, Equation (2) is visually ambiguous as currently typeset, since $v_i$ appears attached to the exponential term; if the intended expression is $e^{q_t k_i^\top} v_i$, I suggest revising the notation to make this unambiguous.

2. **Clarity of the appendix proofs and assumptions on the spectral argument.**
The appendix would benefit from a clearer presentation of the spectral argument for $I-\beta_t(k_t^\top k_t \odot M_t)$. As written, it is not entirely clear what structural assumptions on $M_t$ are required for the stated claims, and whether the entrywise condition $M_t \in [0,1]^{d \times d}$ alone is meant to support the relevant eigenvalue properties. More generally, I think the appendix would be strengthened if the assumptions and conclusions were aligned more explicitly, so that each proof clearly establishes the precise claim being used in the main text.

3. **Practical efficiency trade-off beyond inference-time latency.**
The paper reports that HH-GDN preserves the memory advantage of linear attention but increases inference latency by about $1.4\times$ relative to Gated-DeltaNet, and the limitation section also notes training slowdown due to manual looping when Tensor Cores cannot be used. I believe the paper would be stronger if it reported explicit training-time throughput and memory overhead as a function of rank $r$, since the full practical trade-off is not yet visible from the current efficiency section.

4. **Need for more direct evidence on the claimed mechanism.**
The paper argues that Head-in-Head improves expressiveness by increasing effective rank and diversifying memory interaction patterns. While the entropy-based state-similarity analysis is interesting, it remains somewhat indirect. A more direct analysis of effective rank, learned block interaction structure, or the regimes in which increasing $r$ ceases to help would make the proposed mechanism substantially more convincing.

**Limitations:**

yes

**Strengths And Weaknesses:**

1. **Soundness**
The core idea is technically plausible and supported by fairly broad experiments, but the appendix argument around the spectral behavior of the normalized mask remains insufficiently clear.

2. **Presentation**
The main intuition is understandable, but the paper would benefit from better organization and cleaner exposition, as several tables, equations, and result statements are harder to follow than necessary.

3. **Significance**
The paper studies an important question for linear attention, yet the practical impact is moderated by mostly incremental gains and a reported inference slowdown of about \(1.4\times\) over Gated-DeltaNet.

4. **Originality**
The intra-head grouping idea is novel and gives the paper a distinct contribution, although it is better viewed as a meaningful extension of existing dense-decay designs than as a fundamentally new framework.

---

> ### Author Rebuttal · Authors · 2026-03-28
>
> Thank you for your thorough reading and detailed feedback. We appreciate your positive assessment of the significance and originality of our work.
>
> The core contribution is a general extension method applicable to state transition matrices with off-diagonal elements. By introducing an additional grouping mechanism, we demonstrate consistent improvements across tasks, with acceptable trade-offs in training and inference speed. We will clarify the spectral behavior analysis and revise the layout and tables to improve clarity.
>
> **Question 1: Presentation, Organization of Tables and Figures, and Equations:**
>
> Thank you for pointing these out. We will reorganize figures and tables to enhance readability. In Table 3, the 340M setting should indeed use $L=24$ instead of $L=12$. We will update all tables to consistently highlight the best-performing entries, and revise Equation (2) to resolve the typesetting ambiguity.
>
> **Question 2: Clarifying for the Mask:**
>
> Introducing an arbitrary mask affects eigenvalues, leading to state accumulation during inference. Therefore, we need to regularize the mask to ensure stability of the state transition matrix eigenvalues. Specifically, we require that eigenvalues of $I-\beta_t k_t^\top k_t \odot M$ lie in $[0,1]$, which is equivalent to controlling eigenvalues of $k_t^\top k_t \odot M$ to lie in $[0,1]$, given $\beta_t \in [0,1]$ via sigmoid. We also mention that scaling $\beta_t$ by a factor of 2 allows the eigenvalues of $I-\beta_t k_t^\top k_t \odot M$ to lie in $[-1,1]$.
>
> **Upper bound:** Mask entries all in $[0,1]$ ensure eigenvalues $\leq 1$.
>
> **Lower bound (nonnegative eigenvalues):** Entries in $[0,1]$ alone are insufficient.The supplementary material shows that since $k_t^\top k_t \odot M = \text{Diag}(k_t) M \text{Diag}(k_t)$, requiring the left-hand side eigenvalues to be nonnegative implies $M$ must also have nonnegative eigenvalues:
>
> Proof
> $$
> \forall x,k, xk^\top k\odot M x^{\top} = (x\text{Diag}(k)) M(x\text{Diag}(k))^\top \geq 0 \Rightarrow \forall y, yMy^\top\geq 0
> $$
> Hence, all eigenvalues of $M$ must be nonnegative. Given the arbitrariness of $k$, this condition is both necessary and sufficient.
>
> Our initial approach used Gershgorin circle theorem to enforce diagonal dominance, which improved over baseline but was incompatible with the rank‑1 structure. This motivated our final construction: generate $M_{\text{org}}$ via random $r \times r$ matrices with row-wise L2 normalization and absolute values, then set $M_t = M_{\text{org}}M_{\text{org}}^\top$. This ensures positive eigenvalues bounded by 1, and reverts to rank‑1 when rows of $M_{\text{org}}$ are identical. We will revise the appendix to clarify assumptions and conclusions.
>
>
> The construction can be expressed in code as:
>
> ```python
> ### H for head-number, r for rank
> m = torch.randn([H, r, r])  # can be fixed or generated via nn.Linear(d, H*r*r)
> m_org = l2norm(m.abs()) # norm for the last axis
> m_final = m_org @ m_org.transpose(-1, -2)
> ```
>
>
> **Question 3: Trade-offs and Inference Time:**
>
> We are actively optimizing our training kernels. To fit within A100/A800 shared memory limits, baseline GDN uses chunk size 64, while our method uses chunk sizes 32 for $r=2,4$ and 16 for $r=8$, resulting in comparable or slightly lower memory usage.
>
> Current kernel speeds (single layer, input shape $[B,L,D] = [1,4096,2048]$, 8 heads), 8 heads):
>
> |   | Baseline-GDN | Rank 2| Rank 4| Rank 8|
> | - | - | - | - | -|
> | Memory Cost | 873.3 MB| 797.2 MB | 985.5 MB  | 1.6 GB    |
> | Time Cost   | 6.246 ms     | 8.364 ms | 11.247 ms | 16.078 ms |
>
> Since total training time includes MLP layers of similar duration, the overall training slowdown is less severe than these per-operator numbers suggest. While we reported a $1.4\times$ inference latency in the paper, we believe further kernel optimizations can reduce this gap.
>
> **Question 4: Direct Evidence for the Proposed Mechanism:**
>
> We used state similarity entropy to show that our method encourages the model to learn more diverse representations, demonstrating the effectiveness of this diversity. We also presented results on the effective rank to illustrate the diversity of mask value distributions.
>
> Regarding more direct interpretability, we have not yet found a better approach. Given the inherent difficulty of interpreting large models, linking state values to final performance is challenging, especially since model outputs combine shallow residuals with attention outputs, and deeper layers often revert to identity mappings, rendering states less influential. To meaningfully assess the impact of states on contextual understanding, one would first need to identify which layers and which rows contain effective states. This line of inquiry could lead to a new body of interpretability research, which is beyond the scope of this paper. As for the effect of increasing $r$, we report in review-`w88L` weadkness 1 **Tradeoff for different r**.

---

> > ### Author Rebuttal · Reviewer_o7ec · 2026-04-02
> >
> > 1. Thanks for the extra explanation. I think I understand the intuition better now, but I am still unsure which part of the spectral claim comes from the entrywise constraint alone and which part really requires the construction $M = M_{\mathrm{org}} M_{\mathrm{org}}^\top$. Could you make that distinction more explicit?
> >
> > 2. Thanks also for adding the results with different values of $r$. They are helpful, especially for seeing where the improvement starts to level off. Still, I am not fully convinced that this directly supports the proposed mechanism. Could you say a bit more about what you see as the strongest evidence that the improvement is actually tied to effective-rank increase or block interaction, rather than just extra flexibility in the parameterization?

---

> > > ### Author Response · Authors · 2026-04-02
> > >
> > > Thank you for your reply.
> > >
> > > We apologize that we have still left you with some concerns.
> > >
> > > 1.We reiterate our logic as follows. Due to computational precision issues in long sequences, the eigenvalues of the state transition matrix need to be constrained to the interval [0,1]. Our proof has shown that, under the conditions (where k is L2-normalized and $\beta \in [0,1]$), this can be equivalently transformed into ensuring that the eigenvalues of $M_t$ lie in [0,1]. Any negative factors in $M_t$ can be absorbed into $k_t$, so we only consider the design of $M_t$ within the positive range.
> > >
> > > Based on this, we first ensure that the upper bound holds, which can be simply achieved by constraining all elements of $M_t$ to the interval [0,1]. The proof is provided in the appendix.
> > >
> > > Next, we need to ensure the lower bound holds. One approach is to construct a diagonally dominant matrix, but this cannot naturally revert to a rank-1 structure. Therefore, we generate the matrix by multiplying it with its own transpose. Regardless of the original matrix, this operation guarantees positive eigenvalues and naturally allows a return to the rank-1 structure.
> > >
> > > Based on the above explanation, the final generation scheme we propose is exactly the procedure in the code from my previous reply, which satisfies both requirements simultaneously.
> > >
> > > We acknowledge that there may be other generation schemes that meet these conditions, but they may not be as easy to implement.
> > >
> > > 2.We believe this can be considered a different parameterization. The parameterization method we describe here applies to the entire state transition matrix.
> > >
> > >  At the beginning of the paper, we discussed the gap between the state transition matrix in current linear attention , and a transition matrix where each element is generated completely independently. The main issue is the correlation between off-diagonal elements. The current rank-1 structure causes most elements to be strongly coupled. By introducing our method, the coupling between elements becomes more flexible, leading to better cross-row interaction capabilities.
> > >
> > > Compared to the original DeltaNet and GDN, which generate the transition matrix via a diagonal-plus-rank-1 structure, the additional grouping dimension we introduce increases the rank of cross-row interactions. This distinguishes it from the original rank-1 structure. In practice, the **increased rank is almost equal to the rank of $M_t$ in general**. We can borrow the previous proof:
> > >
> > > $$
> > > rank(\beta_t k_t^{\top}k_t\odot M)=rank(Diag(k_t)M_tDiag(k_t))=rank(M_t) \\
> > > \quad  if k_t = [k_{t1},..k_{tr}],
> > > and \quad k_{ti}  \neq \overrightarrow{0}.  \quad and \quad \beta_t \neq 0.
> > > $$
> > >
> > > Our method can be considered a diagonal-plus-rank-r structure. This constitutes the cross-row interaction pattern and effective rank.
> > >
> > > We conducted another small experiment here. The experimental setup is the same as the visualization mask in the paper. We directly compare the rank of the generation process involving the diagonal elements, i.e., $\beta_t k_t^{\top}k_t\odot M_t$. The computation is performed on sentences of size [B,T]=[10,256]. Since computing the matrix rank in torch requires SVD decomposition, which introduces minor numerical precision issues, we set the tolerance to 0.01. Singular values below this threshold are considered zero and are ignored. This yields the effective rank of $\beta_t k_t^{\top}k_t\odot M \in R^{d\times d}$, and we compare it with the rank of $M_t \in R^{r \times r}$ under the corresponding conditions to verify our previous proof.
> > >
> > > For the average rank computation across all B*T = 2560 matrices, we take the results from one layer as follows:
> > >
> > > |                                    | head1 | head2 | head3 | head4 | head5 | head6 | head7 | head8 |
> > > | ---------------------------------- | ----- | ----- | ----- | ----- | ----- | ----- | ----- | ----- |
> > > | $\beta_t k_t^{\top}k_t\odot M_t$                            | 3.00  | 3.23  | 2.00  | 2.00  | 2.98  | 3.85  | 1.00  | 2.00  |
> > > | $M_t$ | 3     | 4     | 2     | 2     | 3     | 4     | 1     | 2     |
> > >
> > > Next, we simply verify across all $M_t$ and $\beta_t k_t^{\top}k_t\odot M_t$ for L layers and H heads in the entire model, comparing the conjecture that $rank(\beta_t k_t^{\top}k_t\odot M_t)=rank(M)$.
> > >
> > > We simply use the Kolmogorov–Smirnov (KS) distribution test to examine the relationship between the rank distributions of the two. The p-values of mostly layers are close to 1, verifying the previous explanation.
> > >
> > > Our model achieved better results through non rank-1 structures and more cross row interaction modes.
> > >
> > > Hope our response can answer your questions

---

### Decision · Program_Chairs · 2026-04-30

**Decision:**

Accept (regular)

**Comment:**

This paper presents head-in-head linear attention, an approach to enhancing the flexibility of the "decay" transition matrix in linear attention (and state space models). The approach involves a learned masking mechanism that increases the flexibility of the diagonal-plus-low-rank transition matrix in existing DeltaNet-style models. The authors derive a chunkwise parallel form for efficient training, and find that their approach results in improvements against baselines at the 1.3B-param/100B-token scale.

This is an interesting approach to enhancing the flexibility of linear attention models while maintaining hardware-efficient training. The improvements over baselines a clear. While there is some latency overhead, this is primarily a machine learning contribution, and not a kernel contribution---I believe inference could be made much faster with more optimized kernels.

In short, this is a solid contribution to the linear attention architecture space and should be accepted to the conference.